# FARM: Functional Group-Aware Representations for Small Molecules

## Abstract

We introduce Functional Group-Aware Representations for Small Molecules (FARM), a novel foundation model designed to bridge the gap between SMILES, natural language, and molecular graphs. The key innovation of FARM lies in its functional group-aware tokenization, which directly incorporates functional group information into SMILES, enriching SMILES with detailed chemical context. For example, instead of using "O" to represent all oxygen atoms, we use specific tokens like "O_ketone" and "O_hydroxyl" to differentiate oxygen atoms belonging to distinct functional groups. This tokenization expands the chemical lexicon, thereby more effectively bridging SMILES and natural language, ultimately enhancing the model's ability to predict molecular properties. FARM also represents molecules from two perspectives: by using masked language modeling to capture atom-level features and by employing graph neural networks to encode the whole molecule topology. FARM leverages contrastive learning to aligns these two views of representations into a unified molecular embedding. We rigorously evaluate FARM on the MoleculeNet dataset, where it achieves state-of-the-art performance on 11 out of 13 tasks. These results highlight FARM's potential to improve molecular representation learning and demonstrate its strong transfer learning capabilities, paving the way for promising applications in drug discovery and pharmaceutical research.

## 1 Introduction

Artificial intelligence (AI) has emerged as a transformative tool in accelerating scientific discovery, particularly in drug development. It is increasingly employed for tasks such as molecular property prediction, drug-target interaction prediction, and quantitative structure-activity relationship (QSAR) modeling (Chen et al., 2016; Wen et al., 2017; Shen & Nicolaou, 2019; Walters & Barzilay, 2020; Achary, 2020; Wang et al., 2022a; Edwards et al., 2022; Zhang et al., 2023a; Nguyen et al., 2024a; Edwards et al., 2024b;a). However, one of the central challenges in this field is the scarcity of large labeled datasets required for traditional supervised learning methods. This has shifted the focus towards self-supervised pre-trained models that can extract meaningful patterns from vast amounts of unlabeled molecular data (Shen & Nicolaou, 2019). As a result, the development of robust foundation models for molecular representations is now more critical than ever. Despite significant advancements in other domains, such as natural language processing (NLP) and computer vision, there is still no dominant foundation model tailored to molecular representation in drug discovery (Zhang et al., 2023b). This paper begins to address this pressing gap by introducing an innovative approach that leverages functional group (FG)-aware tokenization in the context of both sequence-based and graph-based molecular representations. Analogous to what has been seen with prior studies, aligning these two representations generates a unified and comprehensive molecular embedding that effectively captures both atom-level features and the structural topology of molecules. The innovation here is that the expansion in tokenization granularity in a way that is intentionally interfaced with key drivers of functional properties (i.e., functional groups) enables this form of molecular embedding to better promote models' capacity to understand and predict molecular functions. Figure 1 shows an overview of our molecular representation learning model.

Molecular structures are commonly represented either as sequences, like SMILES or SELFIES, or as molecular graphs. However, relying solely on one type of representation—whether sequence-based or graph-based—limits the ability to capture the full complexity of molecular structures. Sequence

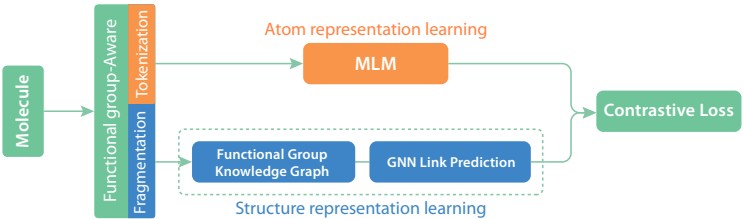

Figure 1: Overview of FARM's molecular representation learning framework.

representations like SMILES can leverage powerful language models, such as BERT (Devlin, 2018) and GPT (Radford, 2018), which have proven highly successful in NLP tasks due to their ability to capture complex patterns, contextual relationships, and semantic nuances in sequential data. However, SMILES strings inherently lose crucial topological information that are often critical for accurate molecular predictions. On the other hand, graph neural networks (GNNs) excel at capturing the local topological structure of molecules but struggle with capturing long-range dependencies, such as interactions between distant atoms (Xu et al., 2018). Our approach overcomes these limitations by integrating the strengths of both representations. We employ masked language models (MLMs) to capture robust atom-level features from SMILES while simultaneously using GNNs to model the structural topology of the molecule. These two representations are aligned through contrastive learning, resulting in a molecular embedding that comprehensively captures both atom-level and structural information. This alignment enables our model to fully represent molecular intricacies, leading to significant improvements in performance across a range of downstream tasks in cheminformatics.

Given that the terms "motifs," "fragments," "substructures," and "building blocks" lack universally accepted definitions in the literature, and their usage varies across different studies, we clarify that functional groups can be considered a subset of these molecular concepts, with a more rigorous definition grounded in chemical principles.

In this work, we integrate functional group information into molecular representations. In the literature, terms such as "motifs," "fragments," "substructures," and "building blocks" are often used interchangeably with "functional groups." However, in this paper, we use the term *"functional groups"* to represent a subset of these molecular concepts, with a more rigorous definition grounded in chemical principles. A functional group refers to a chemically meaningful portion of a molecule that significantly influences its properties and behavior. This can include simple functional groups, such as hydroxyl (-OH), as well as more complex molecular substructures, such as ring systems, which serve distinct functional roles within the molecule. Functional groups (FGs) play a crucial role in determining a molecule's properties, as illustrated in Figure 2. It presents the example of salicylic acid and aspirin, two molecules that share the same core structure but differ by just one functional group. This minor modification has profound implications, leading to large differences in their chemical properties and biological activities.

The key novelty of this work is the introduction of *FG-aware tokenization and fragmentation*, a fine-grained method that enhances molecular representations by incorporating detailed functional group information. This technique applies to both sequence and graph-based models, enriching the molecular representation with chemically meaningful context. The FG-aware tokenization and fragmentation directly incorporate functional group information into the representation of each atom, embedding chemical semantics into the molecular representation. This approach addresses a major limitation of sequence-based models, which typically focus only on individual atom types while neglecting the higher-level functional groups crucial for accurate molecular understanding. FG-aware tokenization enriches the SMILES representation with chemically relevant context, bridging the gap between the expansive vocabularies used in natural language models and the limited chemical lexicon typically available in molecular models, thereby reducing negative transfer in learning. Specifically, to address negative transfer, our FG-aware tokenization expands the vocabulary from 93 tokens to approximately 14,741 tokens by incorporating functional group information. While this significantly increases the complexity of the model, making training slower and harder to converge, it also prevents negative transfer by enabling the model to learn richer and more meaningful chemical semantics. This larger, more nuanced vocabulary allows the model to better capture the

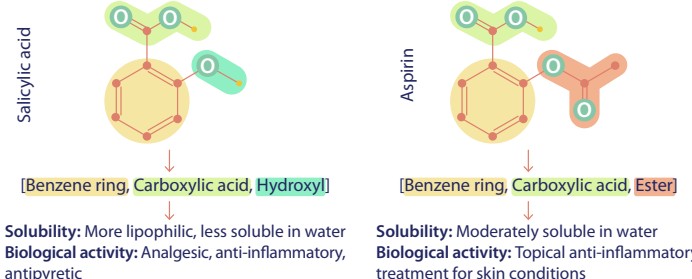

Figure 2: Example of a pair of molecules, salicylic acid and aspirin, that share the same core structure but differ in a single functional group—where the hydroxyl group (-OH) in salicylic acid is replaced by an ester group (-COO-) in aspirin. This small change leads to significant differences in their chemical properties and biological activity.

functional roles of atoms within molecules, improving its ability to generalize across tasks and ultimately leading to more efficient molecular representations.

To further advance the molecular representation, we focus on learning the structural aspects of the molecules. We utilize a *FG knowledge graph* to capture effective embeddings for each functional group, which are then used to learn structural representations of the molecule through link prediction between functional groups. This process ensures that the structural relationships among different functional groups are accurately captured and integrated into the final representation.

By aligning sequence-based and graph-based representations through contrastive loss, our approach achieves state-of-the-art results on 11 out of 13 benchmark tasks in the MoleculeNet dataset (Wu et al., 2018), demonstrating its robustness and versatility. These results underscore the potential of our pre-trained foundation model to significantly advance molecular representation learning, providing a powerful tool for addressing complex challenges in drug discovery and cheminformatics.

In summary, our key contributions include:

- **FG-aware tokenization and fragmentation:** We introduce FG-aware tokenization and fragmentation, adding rich chemical context to each atom and bridging the gap between SMILES and natural language.
- **Structural representation learning:** We leverage a FG knowledge graph for robust FG embeddings and effectively learn molecular structure through link prediction between FGs.
- **Atom-feature and structural representation integration:** We combine masked language model for learning atom-level features with GNNs for capturing structural information.
- **Robustness in downstream tasks:** FARM demonstrates strong transfer learning capabilities - the core goal of pretrained models - outperforming other methods in 11 out of 13 tasks from the MoleculeNet benchmark.

## 2 RELATED WORK

### 2.1 FUNCTIONAL GROUP-AWARE MOLECULAR REPRESENTATIONS

In recent years, there has been growing recognition that incorporating functional group (FG) information into molecular representations can significantly enhance model performance in downstream tasks. Approaches in this domain can be broadly classified into two categories: those leveraging language models (LMs) (Li et al., 2023; Xia et al., 2023) and those utilizing graph neural networks (GNNs) (Zhang et al., 2020; 2021; Yu & Gao, 2022; Yang et al., 2022; Wang et al., 2023; Wu et al., 2023; Han et al., 2023; Fang et al., 2023). Within these categories, methods either employ rule-based functional group detection, relying on predefined chemical rules to identify FGs, or adopt unsupervised strategies that infer substructures or motifs from the data. Regardless of the approach, these models consistently demonstrate that enriching molecular representations with functional group information leads to improved performance across a wide range of molecular property prediction tasks (Fang et al., 2023; Han et al., 2023; Wang et al., 2023). This underlines the importance of functional group awareness in achieving accurate and generalizable molecular representations.

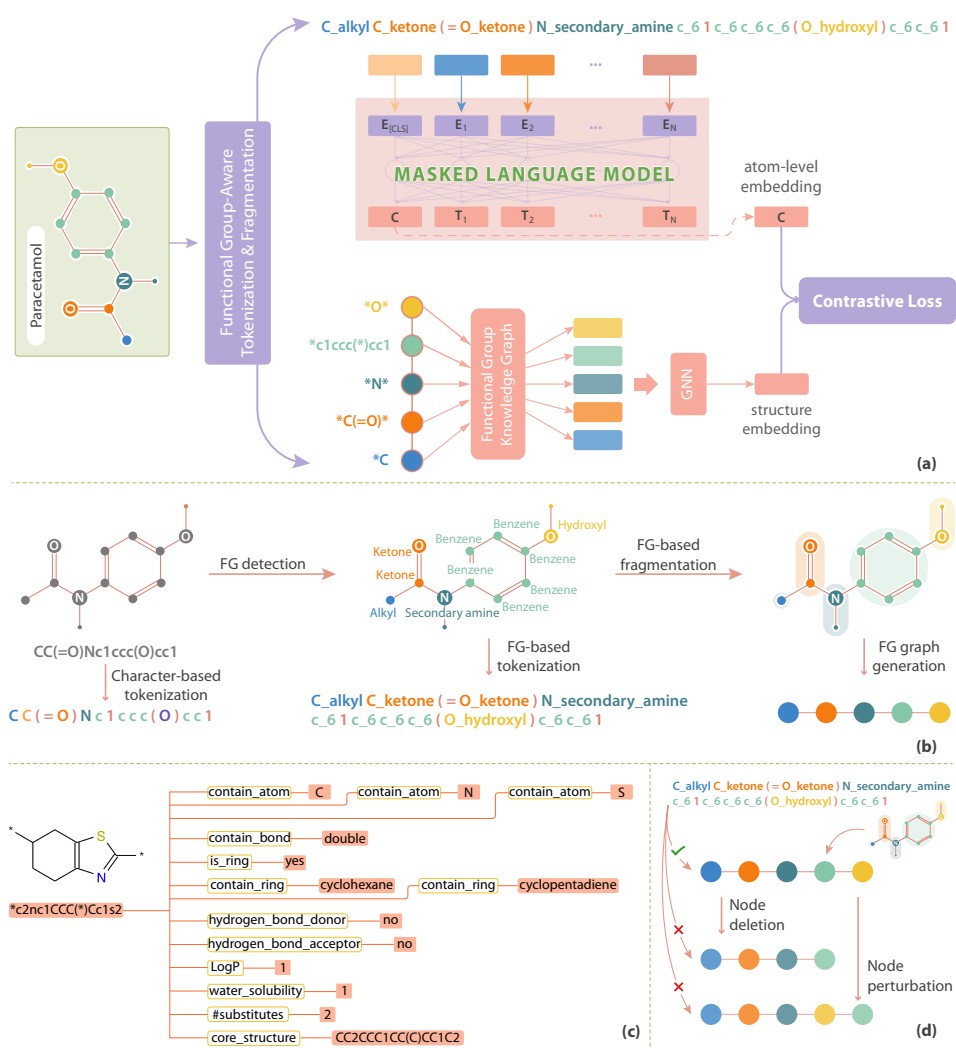

Figure 3: (a) FARM's molecular representation learning model architecture. (b) Functional group-aware tokenization and fragmentation algorithm. (c) Snapshot of the functional group knowledge graph. (d) Generation of negative samples for contrastive learning.

In studies such as Zhang et al. (2021); Han et al. (2023); Chen et al. (2024); Yang et al. (2022), the BRICS algorithm (Degen et al., 2008) is employed to fragment molecules, with some extending the approach through additional rules to achieve finer-grained segmentation. However, BRICS relies on 16 predefined bond-breaking rules rooted in retrosynthetic reactions, which, while useful for synthesis planning, often result in coarse-grained fragmentation. This method focuses on general reaction-based bond breaking, rather than targeting functional group-specific characteristics, and therefore may overlook the detailed structural nuances needed for more precise molecular analysis. Additionally, while BRICS groups atoms into fragments, it does not explicitly identify or label functional groups, limiting its utility in tasks that require functional groups-specific information.

Other works (Li et al., 2023; Chen et al., 2024) leverage RDKit (Landrum, 2010) for functional group detection. While RDKit is effective in identifying common functional groups, its capabilities are limited to well-known groups and do not extend to detecting more complex functional groups, such as ring systems that often dominate molecular datasets and play a critical role in molecular function. Hence, RDKit may overlook less frequent functional groups or larger, more intricate structures that are essential for achieving a comprehensive molecular representation in diverse chemical datasets.

In Wang et al. (2023), the authors propose an unsupervised motif-based graph representation learning technique. This approach can capture local structural motifs but is highly sensitive to the choice

of clustering parameters, and it may struggle to adequately represent more complex ring structures, which are critical in many chemical applications.

In this work, we present a functional group detection algorithm that employs rule-based methods to accurately identify 101 common functional groups, as well as all ring-containing functional groups. Unlike frequent subgraph mining (FSM) and motif-based tokenization, which often yield subgraphs that do not correspond to chemically defined functional groups, our approach is explicitly grounded in chemical principles. By closely aligning with how chemists define functional groups, our method ensures reliable detection of chemically meaningful structures, delivering results that are both accurate and highly relevant to chemistry.

Given a molecule, we can assign each atom to a functional group, ensuring that every atom is associated with a functional unit. This allows us to directly inject functional group information into each atom in SMILES, enriching it with a richer chemical context. By doing so, we can effectively leverage language models (LMs) to learn molecular representations. To the best of our knowledge, we are the first to directly incorporate functional group information into SMILES, providing enhanced chemical context and enabling language models to learn more accurate and meaningful molecular representations.

## 2.2 Contrastive Learning-Based Molecular Representations

Contrastive learning, a self-supervised learning technique, aims to learn representations by maximizing agreement between positive pairs while distinguishing them from negative samples. In the context of molecular representation, Wang et al. (2022b) applies this technique by augmenting each molecular graph to create slightly different versions, which are then treated as negative examples to enhance the learning process. In Pinheiro et al. (2022), the authors use a molecular graph encoder and a SMILES encoder to encode both molecular graphs and SMILES, employing contrastive loss to maximize the agreement between these embeddings. This method enriches SMILES with topology information and the molecular graph with sequence context, making the final embeddings of molecules more robust. In Zhang et al. (2020), the authors minimize the distance between the representation of a molecule and the representations of its constituent substructures. Collectively, these works highlight the versatility of contrastive learning in unifying diverse molecular representations, leading to improved downstream performance in molecular tasks. Our method extends this approach by using contrastive learning to maximize the agreement between FG-enhanced SMILES representations and FG graphs, thereby enhancing the alignment between sequence-based and graph-based representations, ensuring a more comprehensive and chemically informed embedding of molecules.

## 3 Methodology

In this section, we present our methodologies for enhancing molecular representation learning. Section 3.1 introduces FG-aware tokenization and fragmentation, which injects detailed chemical context into both sequence and graph representations. Section 3.2 delves into the masked atom prediction task, a self-supervised technique that enables the model to learn atom-level representations. In Section 3.3, we describe our method for capturing the core molecular structure. Finally, Section 3.4 presents contrastive learning, which aligns FG-enhanced SMILES strings with FG graph embeddings to achieve a comprehensive, unified molecular representation.

## 3.1 Functional group-Aware Tokenization and Fragmentation

We propose an FG-aware tokenization and fragmentation method that embeds detailed functional group information into molecular representations, tailoring it for both SMILES and graph-based models. This method defines a set of functional groups and employs an algorithm to detect them within the molecular graph. The algorithm traverses the graph, evaluating each atom based on criteria such as atom type, neighboring atom types and corresponding bonds, number of neighbors, atom charge, and bonded hydrogen atoms to identify the functional group. This approach ensures precise detection of functional groups within molecular graphs. For instance, a carbon atom with a charge of 0 and three neighbors, including a double-bonded oxygen atom, is classified as part of a ketone group (RCOR'), with the oxygen also contributing to the group. Additionally, we address cases where a functional group may be a subset of another. The algorithm first checks for the

presence of the larger functional group. If it's not identified, the algorithm then checks for smaller functional groups, ensuring correct identification even in complex structures. Appendix B.1 provides a full list of non-ring-containing functional groups.

After identifying conventional functional groups, such as hydroxyl (-OH) and carboxyl (-COOH), the molecule is further analyzed to detect rings and fused ring systems, which are also treated as functional groups. These ring systems are highly diverse and make up a significant portion of all functional groups. For atoms that cannot be assigned to any predefined functional group—typically rare atoms like Ag or Fe—their chemical symbols are used to represent the functional group. This ensures that all atoms are included in the functional group representation, even if they do not fit into standard categories. Once all functional groups are identified, the molecule is segmented at bonds connecting these groups, preserving the functional groups' integrity during fragmentation. The results of the FG detection algorithm are utilized for two distinct processes:

- **FG-aware tokenization:** The molecular graph, where each node (atom) is assigned to a specific functional group, is converted back into a SMILES string that incorporates functional group information. For example, a ketone-containing group originally depicted in SMILES as *C(=O)* is transformed into an FG-enhanced SMILES string like *C_ketone(=O_ketone)*. This FG-enhanced SMILES embeds additional chemical context directly into the molecular representation while remaining fully compliant with traditional SMILES rules (if the FG information is removed, the FG-enhanced SMILES reverts to its standard SMILES form).
- **FG-aware fragmentation:** Once functional groups are identified, the molecule is segmented based on the bonds connecting these groups, as illustrated in Figure 3(b). These bonds are represented as edges in a graph, where the nodes correspond to the functional groups. This structure, known as the functional group graph, conveys the molecule's structural information.

## 3.2 ATOM-LEVEL REPRESENTATION LEARNING

We employ a masked language model architecture that takes FG-enhanced SMILES as input, using atom-level tokenization and leveraging masked atom prediction as a self-supervised task to train the model. Specifically, we adopt the BERT (Devlin, 2018) with a masked language modeling loss:

$$\mathcal{L}_{\text{MLM}} = - \sum_{i \in \mathcal{M}} \log P_\theta(x_i \mid x_{\setminus i}),$$

where $\mathcal{M}$ denotes the set of indices corresponding to the masked tokens, $x_i$ is the original token at position $i$, and $x_{\setminus i}$ represents the input sequence with the token at position $i$ masked. To evaluate the impact of masking, we conduct experiments with varying masking percentages and test the model's performance on six downstream tasks. The results indicate that a masking percentage of 35% yields the highest average performance across tasks. Detailed results are provided in the Appendix D.1.

We examine the attention mechanism of the BERT model trained with FG-enhanced SMILES by visualizing the attention scores for a query atom (Figure 4). The attention map reveals that the model pays more attention to atoms that are strongly connected to the query atom than to those that are merely adjacent in the SMILES string. In detail, the query atom at position 23 shows higher attention to the atom at position 0, which is part of the same ring, rather than to the atom at position 26, which is closer in the SMILES string but not directly connected. This demonstrates that the model effectively learns the syntax and semantics of SMILES, capturing the underlying molecular structure rather than merely relying on the linear sequence of SMILES.

## 3.3 MOLECULAR STRUCTURE LEARNING

To learn the structural view of molecules, we employ a two-step process. First, we derive FG embeddings from the FG knowledge graph. Next, we use these embeddings as inputs for a link prediction model to predict interactions between FGs. This approach effectively captures the relationships among functional groups, both in terms of their structural characteristics and interactions, thereby facilitating the implicit learning of molecular structure.

***FG knowledge graph.*** The FG knowledge graph models each FG with various relations, such as the atoms, bonds, and ring structures it contains, as well as properties like water solubility, lipophilicity (logP), and more. A comprehensive list of FG knowledge graph relations is provided in the Appendix B.1. A snapshot of the FG knowledge graph is shown in Figure 3(c).

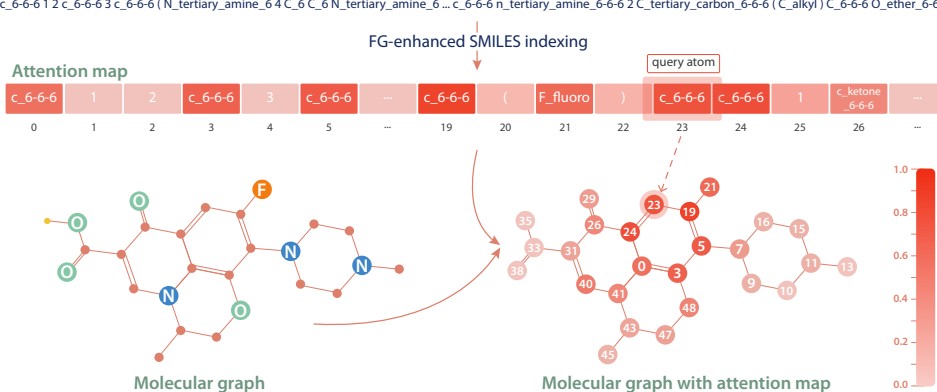

Figure 4: Visualization of the attention map of the BERT model trained with functional group-enhanced SMILES, demonstrating the model's ability to learn and recognize the syntax of SMILES.

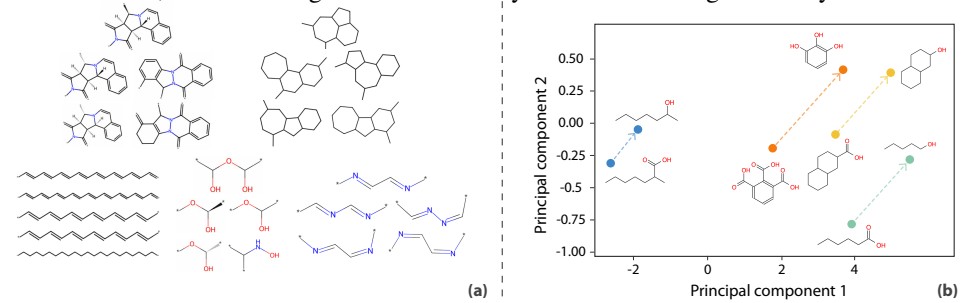

Figure 5: (a) Visualization of functional group knowledge graph embedding space: Clusters of five functional groups with closely related embeddings. (b) Link prediction performance: Substituting one functional group in a molecule with another generates parallel results across different molecules.

The FG knowledge graph embedding is learned by the ComplEx model (Trouillon et al., 2016) to obtain node embeddings. ComplEx is a matrix factorization model specifically designed to learn embeddings from multi-relational data. It is particularly effective at capturing complex, asymmetric relationships and handling one-to-many relations in knowledge graphs, making it well-suited for modeling intricate relational structures. Specifically, each element in a triple $(\mathbf{h}, \mathbf{r}, \mathbf{t})$ — where $\mathbf{h}$ is the head entity, $\mathbf{r}$ is the relation, and $\mathbf{t}$ is the tail entity — is represented as a complex vector. The score for a given triple $(\mathbf{h}, \mathbf{r}, \mathbf{t})$ is calculated as:

$$f(\mathbf{h}, \mathbf{r}, \mathbf{t}) = \operatorname{Re}\left(\mathbf{h}^T \mathbf{r} \cdot \mathbf{t}\right)$$

ComplEx employs a margin-based ranking loss function defined as:

$$\mathcal{L}_{\text{Graph}} = \sum_{(\mathbf{h}, \mathbf{r}, \mathbf{t}) \in E^+} \sum_{(\mathbf{h}', \mathbf{r}, \mathbf{t}') \in E^-} \max\left(0, \gamma + f(\mathbf{h}', \mathbf{r}, \mathbf{t}') - f(\mathbf{h}, \mathbf{r}, \mathbf{t})\right)$$

where $E^+$ denotes the set of positive triples, $E^-$ denotes the set of negative triples, and $\gamma$ represents the margin. Optimizing this loss function will minimize the score of positive triples $(\mathbf{h}, \mathbf{r}, \mathbf{t})$, while maximizing the score of negative triples $(\mathbf{h}', \mathbf{r}, \mathbf{t}')$, with a margin $\gamma$ separating the two. This is achieved through margin-based ranking loss, where the function $f(\mathbf{h}, \mathbf{r}, \mathbf{t})$ evaluates the plausibility of the triples. The goal is to ensure that the score for positive triples is higher than that for negative triples by at least the margin $\gamma$, thus pushing positive triples closer and negative triples farther apart in the embedding space. The detailed implementation of ComplEx is described in Appendix D.2.

By embedding FGs through the knowledge graph, the model can capture both structural and property-based features of each FG, leading to richer FG representations. Figure 5(a) illustrates clusters of FGs in the FG knowledge graph embedding space, showing that similar FGs are closely positioned, suggesting effective structural and property-based grouping.

***Link prediction.*** For link prediction with a graph convolutional network (GCN), we first segment molecules into functional groups through FG-aware molecular segmentation, connecting each group

via single bonds. We then utilize embeddings from the FG knowledge graph as node features for the GCN. The training involves computing node embeddings through graph convolution:

$$\mathbf{h}_i' = \text{ReLU}\left(\mathbf{W} \cdot \frac{1}{|\mathcal{N}(i)|} \sum_{j \in \mathcal{N}(i)} \mathbf{h}_j\right)$$

where $\mathbf{h}_i'$ is the updated embedding for node $i$, computed by averaging the embeddings $\mathbf{h}_j$ of neighboring nodes $\mathcal{N}(i)$, applying the weight matrix $\mathbf{W}$, and passing through ReLU activation function. The score estimates the probability of connections between nodes is computed with a multi-layer perceptron (MLP) and a sigmoid function:

$$p_{ij} = \sigma(\text{MLP}(\mathbf{h}_i \oplus \mathbf{h}_j))$$

We then sample positive edges $E^+$ and negative edges $E^-$, optimizing the model to maximize scores for positive edges and minimize those for negative ones using the loss function:

$$\mathcal{L}_{\text{Link}} = -\frac{1}{|E^+|} \sum_{(i,j) \in E^+} \log p_{ij} - \frac{1}{|E^-|} \sum_{(i,j) \in E^-} \log(1 - p_{ij})$$

Figure 5(b) and Figure 9 (Appendix D.3) demonstrates the capability of our molecular structure representation model, showing that, akin to word pair analogy tasks in NLP, replacing one functional group in a molecule with another (in this case, replacing -OH with -COOH) produces parallel results across different molecules. This demonstrates the model's ability to effectively capture and preserve chemical analogies, highlighting its robustness in learning and representing molecular structures.

## 3.4 Molecular Structure Integration via Contrastive Learning

To integrate FG-enhanced SMILES representations with molecular structure information, we employ contrastive learning to align these representations with FG graph embeddings. This method captures both the atom-level and topological aspects of molecular structures, allowing the model to develop a unified representation that integrates chemical context with overall molecular architecture.

In this framework, each molecule is treated as a pair of representations: the FG-enhanced SMILES and its corresponding FG graph. The contrastive learning task encourages the embeddings of these two representations to be as similar as possible for the same molecule, while pushing apart the representations of different molecules. This allows the model to capture both local chemical features (from the FG-enhanced SMILES) and global molecular topology (from the FG graph).

To enhance the learning process and make it more robust, we generate negative examples by augmenting the FG graph. We apply two types of augmentations: (1) node deletion, where one or more functional groups are removed from the graph, and (2) node swapping, where functional groups are randomly exchanged with one another. Figure 3(d) illustrates how these augmentations are applied to generate negative examples from a FG graph. These augmentations create harder negative examples that force the model to better understand the correct structure and connectivity between functional groups, making it more effective at learning meaningful molecular representations.

Specifically, given a positive pair $(\mathbf{h}_{\text{MLM}}, \mathbf{h}_{\text{pos}})$, where $\mathbf{h}_{\text{MLM}}$ is the atom-level representation derived from a pretrained BERT model and $\mathbf{h}_{\text{pos}}$ is the corresponding structure representation from a graph neural network (GNN), and a negative pair $(\mathbf{h}_{\text{MLM}}, \mathbf{h}_{\text{neg}})$, where $\mathbf{h}_{\text{neg}}$ is a augmented FG-graph, the contrastive loss can be written as:

$$\mathcal{L}_{\text{CL}} = \frac{1}{N} \sum_{i=1}^{N} \max\left(0, \gamma - \text{cosine\_similarity}(\mathbf{h}_{\text{MLM}}, \mathbf{h}_{\text{pos}}) + \text{cosine\_similarity}(\mathbf{h}_{\text{MLM}}, \mathbf{h}_{\text{neg}})\right)$$

where $\gamma$ is the margin parameter and $N$ is the number of training examples (or contrastive pairs). The final objective function for integration is:

$$\mathcal{L}_{\text{Integration}} = \lambda_{\text{MLM}} \cdot \mathcal{L}_{\text{MLM}} + \lambda_{\text{CL}} \cdot \mathcal{L}_{\text{CL}}$$

where $\lambda_{\text{MLM}}$ and $\lambda_{\text{CL}}$ are hyperparameters that control the contribution of each loss to the overall objective. Details on the implementation of contrastive training can be found in the Appendix D.4

Table 1: Evaluation of the FG-enhanced SMILES lexicon size across various databases.

| Database | #molecules | #atom types | Length | $Min\_frequency = 1$ | $Min\_frequency = 5$ |
|---|---|---|---|---|---|
| **ZINC15** | 3M | 10 | [5; 63] | 1,151 | 1,089 |
| **ChEMBL25** | 1.8M | 35 | [1; 867] | 15,269 | 10,016 |
| **Collected dataset** | 20M | 46 | [1; 867] | 22,364 | 14,741 |

## 4 PRE-TRAINING DATA COLLECTION AND DIVERSITY ASSESSMENT

The performance and generalizability of a machine learning model is heavily dependent on the quality and diversity of its training data. Given the vastness of chemical space, it is vital that the training data represents a sufficiently representative subset of this space. Traditional approaches assess the diversity of datasets based on criteria such as the number of chemical elements and the number of atoms per molecule. However, to the best of our knowledge, no previous studies have systematically evaluated the diversity of large chemical datasets like ZINC (Irwin & Shoichet, 2005) or ChEMBL (Gaulton et al., 2011) in terms of functional groups. This gap is significant, as functional groups provide deep insights into the structural and functional complexity of molecules.

In this work, we introduce a novel criterion to assess dataset diversity by using the size of the FG-enhanced SMILES lexicon as a metric. By analyzing the size of this lexicon, we can assess whether the dataset captures a comprehensive range of chemical functionalities, which is crucial for building robust molecular foundation models. Table 1 presents the size of the FG-enhanced SMILES lexicon for the ZINC15 and ChEMBL25 datasets, as well as the dataset we collected for training our foundation model. The table shows that despite its widespread use for training foundation models in small molecule representation, ZINC15 is significantly less diverse compared to ChEMBL25. This is largely due to the fact that ZINC15 is primarily designed to include molecules that adhere to Lipinski's Rule of Five for drug-likeness, excluding more exotic or less common elements that are less relevant to pharmaceutical chemistry. This limited diversity may negatively impact the model's performance on out-of-distribution datasets, which contain a broader range of atom types and functional groups not present in ZINC15. In contrast, ChEMBL25 is far more diverse and thus better suited for training foundation models that can be fine-tuned for various downstream tasks. Based on this insight, we collected a dataset that includes the entire ChEMBL25 database, libraries from chemical drug suppliers, and a subset of ZINC15 to ensure a more comprehensive coverage of chemical space. Detailed information about the collected data can be found in the Appendix A.

## 5 EXPERIMENTAL RESULTS

***Data and splits.*** We consider 12 benchmark tasks in the MoleculeNet dataset (Wu et al., 2018). Following previous works, the data is split using a scaffold split into training, validation, and test sets with an 8:1:1 ratio, ensuring fair and consistent evaluation across all models.

***Evaluation metrics.*** We use ROC-AUC as the evaluation metric for classification tasks due to the high imbalance in some datasets. For physical chemistry tasks (ESOL, Freesolv, and Lipophilicity), we use RMSE, and for quantum mechanics tasks (QM8 and QM9), we use MAE, following previous works. For each downstream task, we split the data using three random seeds, train the models, and report the average and standard deviation of the results.

***Baselines.*** We consider works that incorporate functional group information to enhance representation learning (Zhang et al., 2020; 2021; Yang et al., 2022; Wang et al., 2023; Han et al., 2023; Li et al., 2023; Zang et al., 2023; Wang et al., 2022b; Xia et al., 2023; Nguyen et al., 2024b), as well as other approaches that utilize masked atom prediction as a self-supervised training task (Rong et al., 2020; Hu et al., 2019; Liu et al., 2021; Fang et al., 2022; Zhou et al., 2023).

Table 2 and 3 present the performance of FARM alongside other baseline models on seven MoleculeNet classification and five regression tasks, respectively. FARM consistently outperforms baseline models on various benchmarks, demonstrating its robustness and versatility. Its strong performance across various classification and regression tasks indicates that FARM is a highly effective pretrained model, well-suited for a broad range of downstream tasks in molecular property prediction.

Table 2: Performance comparison of FARM and baseline models on MoleculeNet classification tasks. The first 10 models incorporate FG (functional group) information to enhance representation learning. Performance is evaluated using by ROC-AUC.

| | Physiology | | | | | Biophysics | | |
|---|---|---|---|---|---|---|---|---|
| *Dataset* | **BBBP** | **Tox21** | **ToxCast** | **SIDER** | **ClinTox** | **BACE** | **MUV** | **HIV** |
| *#tasks* | *1* | *12* | *617* | *27* | *2* | *1* | *17* | *1* |
| *#samples* | *2,039* | *7,831* | *8,575* | *1,427* | *1,478* | *1,513* | *93,807* | *41,127* |
| *Evaluation Metric* | *ROC-AUC (%) (↑)* | | | | | *ROC-AUC (%) (↑)* | | |
| **MICRO** (Zhang et al., 2020) | $84.4 \pm 1.1$ | $77.0 \pm 0.8$ | $65.2 \pm 0.8$ | $56.7 \pm 0.9$ | $77.0 \pm 2.0$ | $77.2 \pm 2.0$ | - | $75.1 \pm 1.1$ |
| **MGSSL** (Zhang et al., 2021) | $69.7 \pm 0.9$ | $76.5 \pm 0.3$ | $64.1 \pm 0.7$ | $61.8 \pm 0.8$ | $80.7 \pm 2.1$ | $79.1 \pm 0.9$ | $78.7 \pm 1.5$ | $78.8 \pm 1.2$ |
| **MoleOOD** (Yang et al., 2022) | $71.0 \pm 0.8$ | - | - | $63.4 \pm 0.7$ | - | $84.3 \pm 1.1$ | - | $79.4 \pm 0.5$ |
| **MCM** (Wang et al., 2023) | $90.0 \pm 3.1$ | $80.2 \pm 1.5$ | - | $62.7 \pm 2.8$ | $65.5 \pm 1.4$ | $82.0 \pm 5.5$ | - | - |
| **HimGNN** (Han et al., 2023) | $92.8 \pm 2.7$ | $80.7 \pm 1.7$ | - | $64.2 \pm 2.3$ | $91.7 \pm 3.0$ | $85.6 \pm 3.4$ | - | - |
| **FG-BERT** (Li et al., 2023) | $70.2 \pm 0.9$ | $78.4 \pm 0.8$ | $63.3 \pm 0.8$ | $64.0 \pm 0.7$ | $83.2 \pm 1.6$ | $84.5 \pm 1.5$ | $75.3 \pm 2.4$ | $77.4 \pm 1.0$ |
| **HiMol** (Zang et al., 2023) | $71.3 \pm 0.6$ | $76.0 \pm 0.2$ | - | $62.5 \pm 0.3$ | $70.6 \pm 2.1$ | $84.6 \pm 0.2$ | - | - |
| **Mole-BERT** (Xia et al., 2023) | $71.9 \pm 1.6$ | $76.8 \pm 0.5$ | $62.8 \pm 1.1$ | $62.8 \pm 1.1$ | $78.9 \pm 3.0$ | $80.8 \pm 1.4$ | $78.6 \pm 1.8$ | $78.2 \pm 0.8$ |
| **MolCLR** (Wang et al., 2022b) | $73.3 \pm 1.0$ | $74.1 \pm 5.3$ | - | $61.2 \pm 3.6$ | $89.8 \pm 2.7$ | $82.8 \pm 0.7$ | $78.9 \pm 2.3$ | $77.4 \pm 0.6$ |
| **GLAD** (Nguyen et al., 2024b) | $80.4 \pm 1.5$ | - | - | $64.7 \pm 1.8$ | $87.3 \pm 1.2$ | $85.7 \pm 0.9$ | - | - |
| **N-GRAM** (Hu et al., 2019) | $70.8 \pm 1.5$ | $78.7 \pm 0.4$ | $66.5 \pm 0.3$ | $62.7 \pm 0.8$ | $72.6 \pm 1.5$ | $84.5 \pm 0.7$ | $81.3 \pm 2.1$ | $79.9 \pm 0.7$ |
| **GROVER** (Rong et al., 2020) | $86.8 \pm 2.2$ | $80.3 \pm 2.0$ | $65.3 \pm 0.5$ | $61.2 \pm 2.5$ | $70.3 \pm 13.7$ | $82.4 \pm 3.6$ | $67.3 \pm 1.8$ | $68.2 \pm 1.1$ |
| **GraphMVP** (Liu et al., 2021) | $72.4 \pm 1.6$ | $75.9 \pm 0.5$ | $63.1 \pm 0.4$ | $63.1 \pm 0.4$ | $79.1 \pm 2.8$ | $81.2 \pm 0.9$ | $77.7 \pm 0.6$ | $77.0 \pm 1.2$ |
| **GEM** (Fang et al., 2022) | $88.8 \pm 0.4$ | $78.1 \pm 0.4$ | $69.2 \pm 0.4$ | $63.2 \pm 1.5$ | $90.3 \pm 0.7$ | $87.9 \pm 1.1$ | $75.3 \pm 1.5$ | $81.3 \pm 0.3$ |
| **UniMol** (Zhou et al., 2023) | $72.9 \pm 0.6$ | $79.6 \pm 0.5$ | $69.6 \pm 0.1$ | $65.9 \pm 1.3$ | $\mathbf{91.9 \pm 1.8}$ | $85.7 \pm 0.2$ | $82.1 \pm 1.3$ | $82.8 \pm 0.3$ |
| **FARM** (Ours) | $\mathbf{93.3 \pm 0.2}$ | $\mathbf{80.8 \pm 1.1}$ | $\mathbf{69.9 \pm 0.5}$ | $\mathbf{65.9 \pm 0.7}$ | $82.2 \pm 0.7$ | $\mathbf{89.6 \pm 0.4}$ | $\mathbf{82.7 \pm 2.1}$ | $\mathbf{83.5 \pm 0.5}$ |

Table 3: Performance comparison of FARM and baseline models on MoleculeNet regression tasks. Performance is evaluated by RMSE and MAE.

| | Physical chemistry | | | Quantum mechanics | |
|---|---|---|---|---|---|
| *Dataset* | **ESOL** | **Freesolv** | **Lipophilicity** | **QM8** | **QM9** |
| *#tasks* | *1* | *1* | *1* | *12* | *3* |
| *#samples* | *1,128* | *642* | *4,200* | *21,786* | *133,885* |
| *Evaluation Metric* | *RMSE (↓)* | | | *MAE (↓)* | |
| **HimGNN** (Han et al., 2023) | $0.870 \pm 0.154$ | $1.921 \pm 0.474$ | $0.632 \pm 0.016$ | - | - |
| **FG-BERT** (Li et al., 2023) | $0.944 \pm 0.025$ | - | $0.655 \pm 0.009$ | - | - |
| **Mole-BERT** (Xia et al., 2023) | $1.015 \pm 0.003$ | - | $0.676 \pm 0.002$ | - | - |
| **N-GRAM** (Hu et al., 2019) | $1.100 \pm 0.030$ | $2.510 \pm 0.191$ | $0.880 \pm 0.121$ | $0.0320 \pm 0.003$ | $0.00964 \pm 0.00031$ |
| **GROVER** (Rong et al., 2020) | $1.423 \pm 0.288$ | $2.947 \pm 0.615$ | $0.823 \pm 0.010$ | $0.0182 \pm 0.001$ | $0.00719 \pm 0.00208$ |
| **GEM** (Fang et al., 2022) | $0.813 \pm 0.028$ | $1.748 \pm 0.114$ | $0.674 \pm 0.022$ | $0.0163 \pm 0.001$ | $0.00562 \pm 0.00007$ |
| **MolCLR** (Wang et al., 2022b) | $1.113 \pm 0.023$ | $2.301 \pm 0.247$ | $0.789 \pm 0.009$ | $0.0185 \pm 0.013$ | $0.00480 \pm 0.00003$ |
| **UniMol** (Zhou et al., 2023) | $0.788 \pm 0.029$ | $1.480 \pm 0.048$ | $\mathbf{0.603 \pm 0.010}$ | $0.0156 \pm 0.001$ | $0.00467 \pm 0.00004$ |
| **FARM** (Ours) | $\mathbf{0.761 \pm 0.031}$ | $\mathbf{1.097 \pm 0.033}$ | $0.778 \pm 0.005$ | $\mathbf{0.0146 \pm 0.001}$ | $\mathbf{0.00456 \pm 0.00001}$ |

## 6 CONCLUSION, LIMITATIONS AND FUTURE WORK

In summary, FARM demonstrates robust performance across various MoleculeNet classification tasks, outperforming or matching baseline models. The integration of functional group information and the alignment of FG-enhanced SMILES representations with FG graph embeddings through contrastive learning significantly enhance its effectiveness. This approach underscores FARM's versatility and strength as a pre-trained model, capable of improving molecular structure understanding and predictive accuracy for a wide range of downstream tasks.

While FARM shows strong performance, there are two main limitations that should be addressed in future work. First, the current model does not incorporate a full 3D molecular representation, which is critical for capturing stereochemistry and spatial configurations that affect molecular properties. Incorporating 3D information Yan et al. (2024) could further enhance the model's predictions. Second, the model faces challenges when dealing with rare fused ring systems due to out-of-vocabulary issues. A potential solution to this limitation is to extend the training dataset, covering a broader portion of chemical space to include more diverse and complex molecular structures.

Looking ahead, our ultimate goal is to develop a pre-trained atom embedding that parallels the capabilities of pre-trained word embeddings in NLP. This would enable a richer and more nuanced understanding of molecular properties and behaviors at the atomic level. Similarly, we aim to achieve

molecule-level representations that are as expressive and versatile as sentence-level embeddings in NLP, capturing both local and global molecular features. By bridging the gap between atom-wise embeddings and holistic molecule representations, FARM paves the way for more accurate, generalizable molecular predictions across a variety of tasks.

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

## A  MOLECULAR DATASETS

### A.1  TRAINING DATA

We collected a diverse dataset to train our **FARM** model from various sources, including ChEMBL25, ZINC15, and several chemical suppliers. The number of compounds in each dataset is reported as follows:

Table 4: List of compound suppliers and number of compounds

| Supplier | Number of Compounds | Source |
|---|---|---|
| Targetmol | 22,555 | https://www.targetmol.com/ |
| Chemdiv | 1,741,620 | https://www.chemdiv.com/ |
| Enamine | 862,698 | https://enamine.net/ |
| Life Chemical | 347,657 | https://lifechemicals.com/ |
| Chembridge | 1,405,499 | https://chembridge.com/ |
| Vitas-M | 1,430,135 | https://vitasmlab.biz/ |
| InterBioScreen | 560,564 | https://www.ibscreen.com/ |
| Maybridge | 97,367 | https://chembridge.com/ |
| Asinex | 601,936 | https://www.asinex.com/ |
| Eximed | 61,281 | https://eximedlab.com/ |
| Princeton BioMolecular | 1,647,078 | https://princetonbio.com/ |
| Otava | 9,203,151 | https://www.otava.com/ |
| Alinda Chemical | 733,152 | https://www.alinda.ru/synthes_en.html |
| ChEMBL 25 | 1,785,415 | https://www.ebi.ac.uk/chembl/ |
| ZINC15 | 4,000,000 | https://zinc15.docking.org/ |
| **Total** | 20,000,000 | |

### A.2  DOWNSTREAM TASKS DATA

In Table 5, we provide an overview of the datasets used for evaluating the performance of our model on various downstream tasks. Each dataset is denoted by its name, followed by the number of tasks it encompasses, the total number of samples available in each dataset, and a brief description. These datasets cover a range of chemical and biological properties, enabling comprehensive evaluation of the model's performance across different tasks in molecular representation learning.

## B  FG-AWARE TOKENIZATION AND FRAGMENTATION

### B.1  THE LIST OF FUNCTIONAL GROUPS

The exhaustive list of 101 functional groups that can be detected by the functional group detection algorithm includes: Tertiary carbon, Quaternary carbon, Alkene carbon, Cyanate, Isocyanate, Hydroxyl, Ether, Hydroperoxy, Peroxy, Haloformyl, Aldehyde, Ketone, Carboxylate, Carboxyl, Ester, Hemiacetal, Acetal, Hemiketal, Ketal, Orthoester, Carbonate ester, Orthocarbonate ester, Amidine, Carbamate, Isothiocyanate, Thioketone, Thial, Carbothioic S-acid, Carbothioic O-acid, Thiolester, Thionoester, Carbodithioic acid, Carbodithio, Trifluoromethyl, Difluorochloromethyl, Bromodifluoromethyl, Trichloromethyl, Bromodichloromethyl, Tribromomethyl, Dibromofluoromethyl, Triiodomethyl, Difluoromethyl, Fluorochloromethyl, Dichloromethyl, Chlorobromomethyl, Chloroiodomethyl, Dibromomethyl, Bromoiodomethyl, Diiodomethyl, Alkyl, Alkene, Alkyne, Carboxylic anhydride, Primary amine, Secondary amine, Amide, Imide, Tertiary amine, 4-ammonium ion, Hydrazone, Primary ketimine, Primary aldimine, Secondary ketimine, Secondary aldimine, Nitrile, Azide, Azo, Nitrate, Isonitrile, Nitrosooxy, Nitro, Nitroso, Aldoxime, Ketoxime, Sulfhydryl, Sulfide, Disulfide, Sulfinyl, Sulfonyl, Sulfur dioxide, Sulfuric acid, Sulfino, Sulfonic acid, Sulfonate ester, Thiocyanate, Phosphino, Phosphono, Phosphate, Phosphodiester, Phosphoryl, Borono, Boronate, Borino, Borinate, Silyl ether, Dichlorosilane, Trimethylsilyl, Fluoro, Chloro, Bromo, Iod.

Table 5: Overview of downstream tasks, corresponding sample sizes, and dataset descriptions.

| Dataset | # Tasks | # Samples | Description |
|---|---|---|---|
| BBBP | 1 | 2,039 | Benchmark for Blood-Brain Barrier permeability prediction, assessing whether compounds can cross the blood-brain barrier. |
| Tox21 | 12 | 7,831 | Toxicology data containing multiple assays for evaluating the toxicity of compounds across various endpoints. |
| SIDER | 27 | 1,427 | Side Effect Resource dataset that includes drug side effects associated with FDA-approved drugs, focusing on adverse drug reactions. |
| ClinTox | 2 | 1,478 | Clinical Toxicology dataset designed to predict the toxicity of drug-like compounds based on clinical data. |
| BACE | 1 | 1,513 | Data for predicting activity against the beta-secretase enzyme, relevant for Alzheimer's disease drug discovery. |
| MUV | 17 | 93,807 | Multiple Unrelated Variables dataset aimed at assessing the ability to predict various molecular properties and activities. |
| HIV | 1 | 41,127 | Dataset focused on predicting the activity of compounds against the HIV virus, crucial for antiviral drug development. |
| ESOL | 1 | 1,128 | Dataset used for estimating the solubility of organic compounds in water, useful for understanding compound behavior in biological systems. |
| FreeSolv | 1 | 642 | Dataset containing free energy of solvation values for small organic molecules in water, aiding in solvation energy predictions. |
| Lipophilicity | 1 | 4,200 | Data focused on predicting the octanol-water partition coefficient, a key measure of a compound's lipophilicity. |
| QM8 | 12 | 21,786 | Quantum Mechanics dataset that provides a range of molecular properties computed using quantum mechanical methods for small organic molecules. |
| QM9 | 3 | 133,885 | Quantum Mechanics dataset providing molecular properties for a large set of small organic compounds. |

## B.2 NAMING FUNCTIONAL GROUPS WITH RINGS IN FUSED RING SYSTEMS

Fused ring systems are a diverse and prevalent class of functional groups, accounting for 99.37% of the total functional groups in our dataset (147,564 out of 148,507 FGs). Despite their importance, many of these systems lack standardized nomenclature. To address this, we propose a systematic approach to naming these ring systems based on their ring sizes and core structures.

Each ring in a fused ring system is named according to its size. For instance, a six-membered aromatic ring like benzene is named ring_6. This straightforward approach provides a clear identifier for individual rings within a system. For systems composed of multiple fused rings, we use the following naming convention:

- **Identification:** Determine the smallest atom index for each ring within the system.
- **Sorting:** Arrange the rings by increasing atom indices.
- **Construction:** Combine the ring sizes in ascending order. For example, a fused system with a six-membered ring and a five-membered ring would be named ring_5_6.

This systematic naming helps in identifying and categorizing complex fused ring systems by focusing on their core structure. The core structure is defined as the central framework of interconnected rings that forms the fundamental backbone of the molecule. The core structure of a ring system is important because it influences the molecule's reactivity, stability, and biological activity. In SMILES notation, which uses lowercase characters to indicate atoms within aromatic rings, we can enhance the representation by combining the atom symbol (uppercase or lowercase) with the core structure, thereby providing a comprehensive depiction of the ring system. Figure 6a illustrates an example of naming a fused ring system based on the rules described above, and Figure 6b shows how FG-aware tokenization is applied.

After completing the naming process, we derive a new FG-enhanced SMILES representation for the molecules. We then analyze our collected dataset, which comprises 20 million samples of FG-enhanced SMILES, to evaluate the results. This dataset includes representations of 46 different elements. Notably, 11 elements are represented by only a single form, indicating their rare occurrence within the dataset (excluding hydrogen). These elements are: H, Ti, V, Cr, Rb, Mo, Rh, Sb, Ba, Pb, and Bi. In contrast, the remaining 35 elements feature at least two representations, each cor-

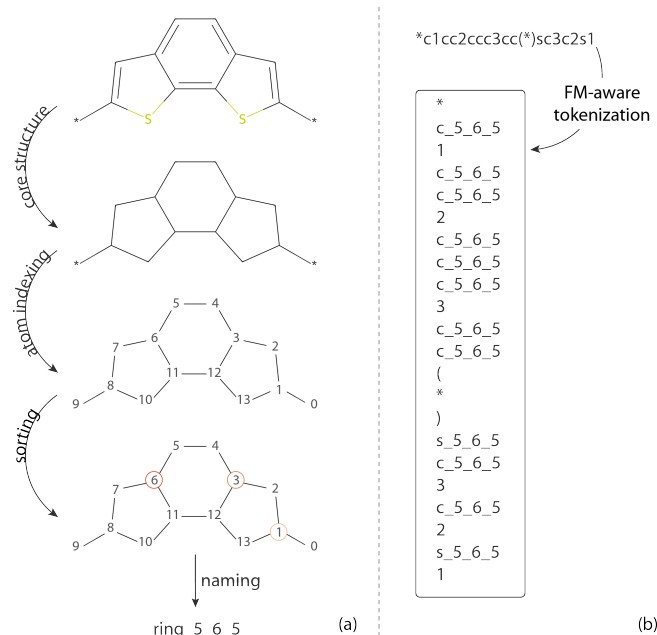

Figure 6: (a) Example of naming a fused ring system in 4 steps: generate the core structure of the functional group, index atoms using RDKit, select the smallest-index atom in each ring and sort, and name the fused ring system based on ring size. (b) Example of FG-aware tokenization.

responding to distinct FGs. The distribution of these elements is visualized in Figure 7, highlighting the diversity of representations in our dataset. The most prevalent element in our dataset is Carbon, with 9,112 FGs containing it. Nitrogen follows as the second most prevalent element, represented in 2,549 FGs, while Oxygen and Sulfur appear in 2,156 and 571 FGs, respectively.

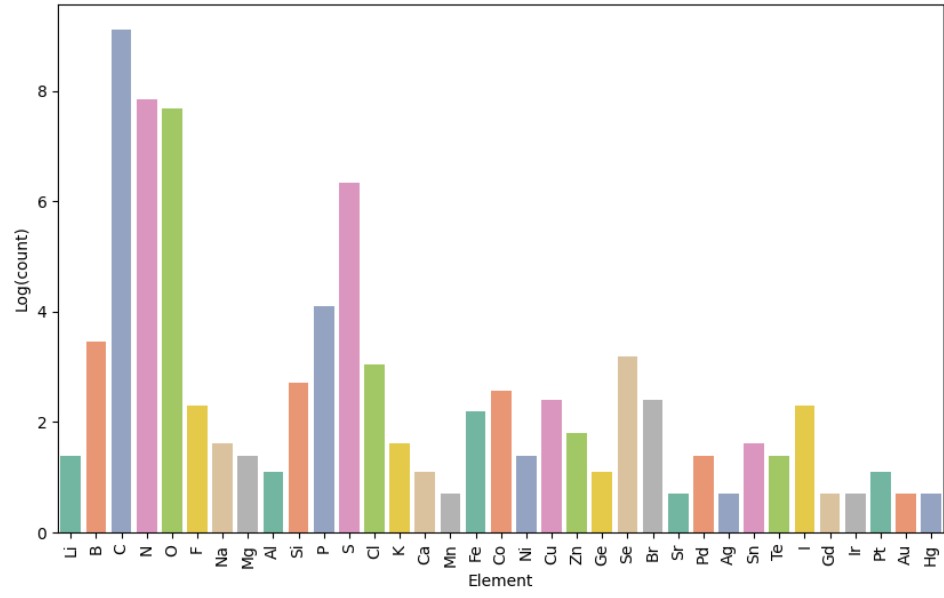

Figure 7: Number of functional groups associated with different chemical elements in the FG-enhanced SMILES dataset. The y-axis represents the natural logarithm (log, base $e$) of the count.

## C   FG KNOWLEDGE GRAPH

The FG knowledge graph is designed to capture both the structural and property-related information of FGs. The list of relations includes:

Table 6: Key relations defined in the FG knowledge graph. (Note: Continuous values, such as LogP and water solubility, are discretized by rounding to the nearest integer.)

| Relation | Description |
|---|---|
| contain_atom | Identifies atoms present in the FG (e.g., C, H, O, N). |
| contain_bond | Specifies types of bonds in the FG (e.g., single, double, triple, aromatic). |
| functional_group | Recognizes functional groups in the FG (e.g., hydroxyl, carboxyl, amine). |
| contain_ring_[n] | Indicates the presence of a non-aromatic ring of size n in the FG. |
| contain_aromatic_ring_[n] | Indicates the presence of an aromatic ring of size n in the FG. |
| num_substitutes | Specifies the number of substituents (e.g., alkyl or aryl groups) in the FG. |
| is_hydrogen_bond_donor | Identifies whether the FG contains a functional group capable of donating hydrogen bonds. |
| is_hydrogen_bond_acceptor | Identifies whether the FG contains a functional group capable of accepting hydrogen bonds. |
| logp | Measures the lipophilicity of the FG using the logP value (calculated via RDKit). In the collected dataset, values range from -35 to 31. |
| water_solubility | Predicts the solubility of the FG in water, based on logP, molecular weight, and TPSA. In the collected dataset, values range from -5 to 8. |
| core_smiles | The SMILES representation of the core structure of the FG. |

- **List of functional groups that act as hydrogen bond donors:** Hydroxyl, Hydroperoxy, Primary amine, Secondary amine, Hydrazone, Primary ketimine, Secondary ketimine, Primary aldimine, Amide, Sulfhydryl, Sulfonic acid, Thiolester, Hemiacetal, Hemiketal, Carboxyl, Aldoxime, Ketoxim.

- **List of functional groups that act as hydrogen bond acceptors:** Ether, Peroxy, Haloformyl, Ketone, Aldehyde, Carboxylate, Carboxyl, Ester, Ketal, Carbonate ester, Carboxylic anhydride, Primary amine, Secondary amine, Tertiary amine, 4-Ammonium ion, Hydrazone, Primary ketimine, Secondary ketimine, Primary aldimine, Amide, Sulfhydryl, Sulfonic acid, Thiolester, Aldoxime, Ketoxi.

## D   IMPLEMENTATION DETAILS

### D.1   TRAINING MASKED LANGUAGE MODEL FOR SMILES REPRESENTATION

We trained the BERT model using Hugging Face (Wolf et al., 2020) on the masked molecule prediction task with both conventional SMILES and FG-enhanced SMILES from our collected dataset. To assess the impact of different masking percentages, we trained BERT models with masking percentages of 0.15, 0.25, 0.35, 0.45, and 0.55. The models were then evaluated on seven MoleculeNet tasks, including three classification tasks and four regression tasks, to determine the optimal masking percentage. The results, presented in Table 7, indicate that a masking percentage of 0.35 yields the best performance across the considered downstream tasks.

Table 7: Performance of BERT models with varying masking percentages across six MoleculeNet tasks. The data is split using a random split into training, validation, and test sets with an 8:1:1 ratio.

| | BBBP | BACE | HIV | Average | ESOL | FreeSolv | Average | QM9 |
|---|---|---|---|---|---|---|---|---|
| #tasks | 1 | 1 | 1 | | 1 | 1 | | 3 |
| #samples | 2039 | 1513 | 41127 | | 1128 | 642 | | 133885 |
| Metric | | ROC-AUC (↑) | | | | RMSE (↓) | | MAE (↓) |
| 0.25 | 93.01 ± 0.9 | 94.31 ± 1.08 | 80.17 ± 1.5 | 89.16 | 0.688 ± 0.033 | 0.622 ± 0.007 | 0.655 | 0.0091 ± 0.00001 |
| 0.25 | 93.59 ± 1.7 | 93.94 ± 1.4 | 81.03 ± 1.9 | 89.52 | 0.543 ± 0.030 | 0.714 ± 0.010 | 0.629 | **0.0032 ± 0.00001** |
| 0.35 | 94.36 ± 0.5 | 94.54 ± 0.4 | 81.93 ± 1.7 | **90.27** | 0.608 ± 0.031 | 0.507 ± 0.030 | **0.558** | 0.0041 ± 0.00001 |
| 0.45 | 93.48 ± 1.3 | 94.36 ± 0.90 | 80.12 ± 1.7 | 89.32 | 0.795 ± 0.028 | 0.493 ± 0.008 | 0.644 | 0.0048 ± 0.00001 |
| 0.55 | 92.85 ± 1.1 | 88.68 ± 1.0 | 79.89 ± 0.90 | 87.14 | 0.734 ± 0.030 | 0.599 ± 0.005 | 0.667 | 0.0097 ± 0.00001 |

Additional details of the training setup include training the BERT model on 20 million SMILES for 15 epochs using two NVIDIA Tesla V100 GPUs. The learning rate was set to $1e-5$, with a batch size of 128, and model checkpoints were saved after every 10,000 batches. This setup was also applied to the baseline model, which used conventional SMILES for comparison.

Figure 8 illustrates the convergence behavior of the models trained on different representations of molecular data. The model utilizing FG-enhanced SMILES exhibits a slower convergence rate, attributed to the increased complexity of its vocabulary, reflecting its closer resemblance to natural language. The SMILES model converges by step 200 (after processing 25,600 SMILES), while the FG-enhanced SMILES model achieves convergence by step 300 (after processing 38,400 SMILES). Notably, despite the larger prediction vocabulary (14,714 vs. 93), the FG-enhanced model ultimately reaches a lower loss, suggesting its enhanced capacity to capture intricate molecular representations and improve generalization in complex tasks. This indicates the model's ability to leverage functional group information effectively, potentially leading to better performance in downstream applications.

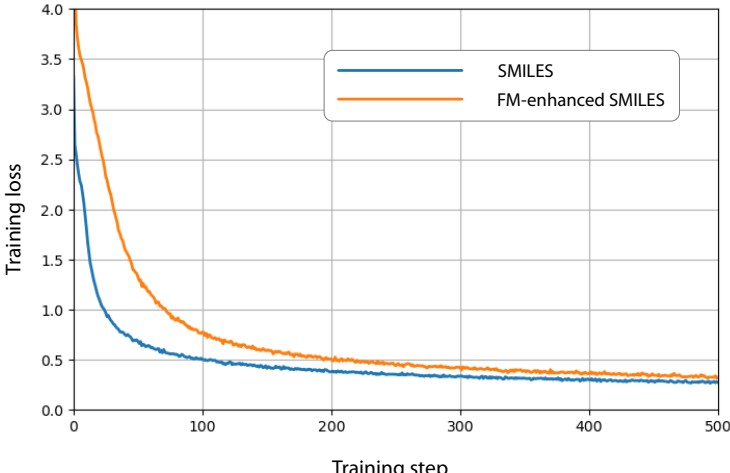

Figure 8: Loss curves for the masked language model (MLM) during training on two datasets: standard SMILES and functional group-enhanced SMILES.

### D.2 TRAINING FG KNOWLEDGE GRAPH EMBEDDING MODEL FOR MOLECULAR STRUCTURE REPRESENTATION

Once the FG knowledge graph is constructed as detailed in Section C, we utilize the ComplEx model to learn embeddings for the functional groups. The knowledge graph comprises 148,507 unique nodes: 147,564 corresponding to ring systems and 943 representing non-ring functional groups. Training is conducted with a batch size of 64, a learning rate of $1 \times 10^{-3}$, over 50 epochs, with model checkpoints saved at the end of each epoch.

**ComplEx Model Representation**

In the ComplEx model (Trouillon et al., 2016), each element in a triple $(\mathbf{h}, \mathbf{r}, \mathbf{t})$ — where $\mathbf{h}$ is the head entity, $\mathbf{r}$ is the relation, and $\mathbf{t}$ is the tail entity — is represented as a complex vector:

$$\mathbf{h}, \mathbf{r}, \mathbf{t} \in \mathbb{C}^d \tag{1}$$

*Scoring Function*

The score for a given triple $(\mathbf{h}, \mathbf{r}, \mathbf{t})$ is calculated as:

$$f(\mathbf{h}, \mathbf{r}, \mathbf{t}) = \text{Re}\left(\mathbf{h}^T \mathbf{r} \cdot \mathbf{t}\right) \tag{2}$$

where $\mathbf{r}$ is a complex-valued vector, and the dot product is performed in the complex space.

*Loss Function*

ComplEx employs a margin-based ranking loss function defined as:

$$\mathcal{L}_{\text{Graph}} = \sum_{(h,r,t) \in E^+} \sum_{(\mathbf{h}',\mathbf{r},\mathbf{t}') \in E^-} \max\left(0, \gamma + f(\mathbf{h}', \mathbf{r}, \mathbf{t}') - f(\mathbf{h}, \mathbf{r}, \mathbf{t})\right) \tag{3}$$

where $E^+$ denotes the set of positive triples, $E^-$ denotes the set of negative triples, and $\gamma$ represents the margin.

To assess the quality of the learned embeddings, we randomly sample clusters of five closely related embedding vectors and analyze their arrangement in the embedding space. The results of this evaluation are presented in Figure 5a.

### D.3 LINK PREDICTION MODEL USING GNNS

For link prediction using the GCN model, we start by segmenting molecules into functional groups via FG-aware molecular segmentation, where each group is connected by single bonds. We then use embeddings from the FG knowledge graph embedding model as node features for the GCN. The training process involves computing node embeddings through graph convolution (Equation 4), followed by scoring potential edges with a multi-layer perceptron (MLP) (Equation 5). This score is used to calculate the probability between two nodes (Equation 6). Positive and negative edges are sampled, and the model is optimized to maximize scores for positive edges while minimizing scores for negative edges using the loss function in Equation 7. This approach effectively trains the model to distinguish between likely and unlikely connections between functional groups.

$$\mathbf{h}'_i = \text{ReLU}\left(\mathbf{W} \cdot \frac{1}{|\mathcal{N}(i)|} \sum_{j \in \mathcal{N}(i)} \mathbf{h}_j\right) \tag{4}$$

where $\mathbf{h}'_i$ is the updated embedding for node $i$. It is computed by averaging the embeddings $\mathbf{h}_j$ of neighboring nodes $\mathcal{N}(i)$, applying the weight matrix $\mathbf{W}$, and then passing through the ReLU activation function.

$$s_{ij} = \text{MLP}(\mathbf{h}_i \oplus \mathbf{h}_j) \tag{5}$$

where $s_{ij}$ denotes the score assigned to the potential edge between nodes $i$ and $j$. The score is computed using a multi-layer perceptron (MLP), which takes as input the concatenated node embeddings of $i$ and $j$, denoted as $\mathbf{h}_i \oplus \mathbf{h}_j$. Here, $\mathbf{h}_i$ and $\mathbf{h}_j$ represent the node embeddings for nodes $i$ and $j$, respectively. The operator $\oplus$ indicates the concatenation of these embeddings. The MLP processes this concatenated vector to produce a score that reflects the likelihood of an edge existing between $i$ and $j$.

$$p_{ij} = \sigma(s_{ij}) \tag{6}$$

where $\sigma$ is the sigmoid function.

$$\mathcal{L}_{\text{Link}} = -\frac{1}{|E^+|} \sum_{(i,j) \in E^+} \log p_{ij} - \frac{1}{|E^-|} \sum_{(i,j) \in E^-} \log(1 - p_{ij}) \tag{7}$$

where $\mathcal{L}$ is the loss function for link prediction. It computes the average log-likelihood of positive edges $E^+$ and negative edges $E^-$, where $p_{ij}$ is the predicted probability of an edge between nodes $i$ and $j$. The loss penalizes the model for incorrect predictions, encouraging high probabilities for true edges and low probabilities for false edges.

The GCN model for link prediction is trained as follows: For each molecule, represented as a FG graph, we generate all possible combinations of nodes, encompassing both positive pairs (nodes that are linked) and negative pairs (nodes that are not linked). In cases where the graph contains more than three nodes (FGs), we select 60% of all possible combinations along with all positive pairs to form the training data for each graph. The model is subsequently trained for three epochs on a comprehensive dataset consisting of 20 million data points. Figure 9 shows the performance of the link prediction model. Similar to word embedding analogies in NLP, replacing one FG in a molecule

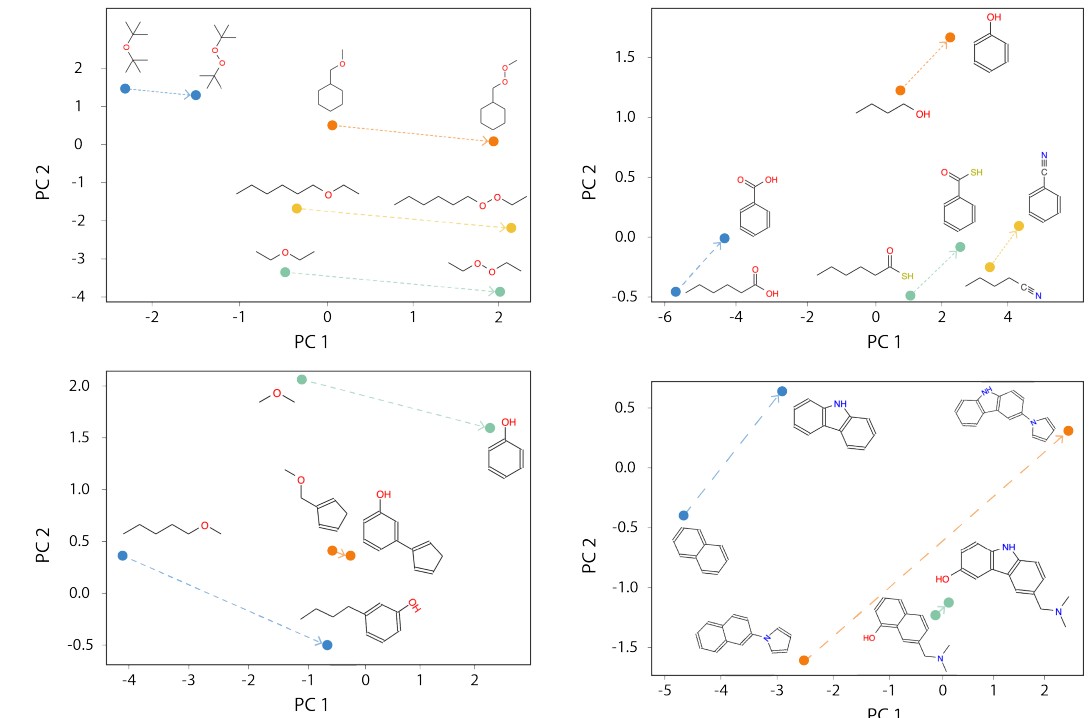

Figure 9: Link prediction model performance: Similar to word embedding analogies in NLP, replacing one functional group in a molecule with another produces parallel results across different molecules, demonstrating the model's ability to capture chemical relationships effectively.

with another produces parallel results across different molecules, demonstrating the model's ability to capture chemical relationships effectively.

### D.4 CONTRASTIVE LEARNING: ALIGN SMILES AND STRUCTURE REPRESENTATION

In this work, we propose a contrastive learning strategy to align SMILES-based representations of molecules with their corresponding graph-based molecular structures. The goal of this approach is to capture both the sequential information from SMILES and the structural relationships encoded in graph representations, thus allowing the model to learn a more comprehensive molecular representation that bridges these two modalities.

To measure the similarity between representations derived from the FG-enhanced SMILES and FG graph, we utilize cosine similarity, which is defined as: The cosine similarity between two vectors $\mathbf{u}$ and $\mathbf{v}$ is defined as:

$$\text{cosine\_similarity}(\mathbf{u}, \mathbf{v}) = \frac{\mathbf{u} \cdot \mathbf{v}}{\|\mathbf{u}\| \|\mathbf{v}\|}$$

Here, $\mathbf{u}$ and $\mathbf{v}$ represent the embeddings from two different modalities, such as the SMILES-based BERT output and the GNN output for the molecular graph. This similarity score helps ensure that embeddings of positive (i.e., matched) SMILES and graph representations are closer in the latent space.

To align these two types of representations, we use contrastive loss, a popular technique in self-supervised learning that enforces representations from the same sample (positive pair) to be more similar than those from different samples (negative pair). Given a positive pair $(\mathbf{h}_{\text{MLM}}, \mathbf{h}_{\text{pos}})$, where $\mathbf{h}_{\text{MLM}}$ is the SMILES representation derived from a pretrained BERT model and $\mathbf{h}_{\text{pos}}$ is the corresponding representation from a graph neural network (GNN), and a negative pair $(\mathbf{h}_{\text{MLM}}, \mathbf{h}_{\text{neg}})$, where $\mathbf{h}_{\text{neg}}$ is a augmented FG-graph, the contrastive loss can be written as:

$$\mathcal{L}_{\text{CL}} = \frac{1}{N} \sum_{i=1}^{N} \max\left(0, \gamma - \text{cosine\_similarity}(\mathbf{h}_{\text{MLM}}, \mathbf{h}_{\text{pos}}) + \text{cosine\_similarity}(\mathbf{h}_{\text{MLM}}, \mathbf{h}_{\text{neg}})\right)$$

Where:

- $\gamma$ is the margin parameter, ensuring that the positive similarity is significantly larger than the negative similarity.
- $N$ is the number of training examples (or contrastive pairs)

The objective function is

$$\mathcal{L} = \lambda_{\text{MLM}} \cdot \mathcal{L}_{\text{MLM}} + \lambda_{\text{CL}} \cdot \mathcal{L}_{\text{CL}}$$

where $\mathcal{L}_{\text{MLM}}$ represents the masked language modeling loss, which encourages the model to predict masked tokens in the input sequence effectively, and $\mathcal{L}_{\text{CL}}$ denotes the contrastive loss, which aligns the SMILES and structural representations. The coefficients $\lambda_{\text{MLM}}$ and $\lambda_{\text{CL}}$ are hyperparameters that control the contribution of each loss to the overall objective. By tuning these coefficients, we can balance the learning process between the two tasks, allowing the model to learn rich and meaningful representations from both the sequential and structural aspects of the molecular data.

This combined loss function enables the model to leverage the strengths of both masked language modeling and contrastive learning, fostering a more comprehensive understanding of molecular representations that can enhance performance in downstream tasks such as property prediction, molecular generation, and structure-based drug discovery.

In our contrastive learning model, we set the margin $\gamma = 0.5$ and the weights $\lambda_{MLM} = 1.0$ and $\lambda_{CL} = 0.5$. We train the contrastive BERT model using a batch size of 126 for a total of 5 epochs. This training configuration mirrors the setup used for learning atom representations with the BERT model, as described in Section D.1.

### D.5 Downstream task finetuning

MoleculeNet tasks are treated as downstream tasks for our **FARM** model. We freeze all layers of FARM and pair it with a GRU head for both classification and regression tasks. For classification, we use cross-entropy as the loss function, while for regression, we employ mean squared error. The Adam optimizer is applied with a learning rate of $1e-4$ and a cosine annealing learning rate schedule with a period of 20 epochs. The training process spans 100 epochs with a batch size of 16, using an 80-10-10 train-validation-test split with scaffold splitting. To address imbalanced datasets, we implement a weighted loss function, assigning a weight of 5 to classes with fewer samples. For each task, we conduct three runs with different train-validation-test splits and report the average and standard deviation of the results.

## E  Ablation Study

To assess the effectiveness of each component in our architecture, we conducted a comprehensive ablation study across several MoleculeNet benchmark tasks. The first model, FM_KGE + GAT, utilizes FG knowledge graph embeddings as input for a Graph Attention Network (Veličković et al., 2017) (GAT) to predict molecular properties. Although its performance on these tasks is not the strongest, the model still demonstrates its capacity to learn underlying chemical rules (syntax and semantics) from the data to a certain degree.

The second model, AttentiveFP (Xiong et al., 2019), performs a masked atom prediction task on the molecular graph, predicting atom types such as carbon, hydrogen, oxygen, and nitrogen. Its variation, FG AttentiveFP, shares the same architecture as AttentiveFP, but it predicts both the atom type and the associated functional group. Experimental results indicate that incorporating functional group information significantly improves the model's performance on downstream tasks.

We also evaluate the BERT model trained on canonical SMILES strings, and its counterpart, FG BERT, which is trained on FG-enhanced SMILES. Results show that providing additional chemical context about functional groups boosts model performance in downstream tasks.

Finally, **FARM** (FG BERT with contrastive learning) integrates molecular structure representations from link prediction embeddings. **FARM** consistently achieves the highest performance across 6 out of 7 downstream tasks, demonstrating the power of combining FG-enhanced SMILES and contrastive learning.

Table 8 presents the detailed results of the aforementioned models across various MoleculeNet tasks, illustrating the performance of each architecture. For these experiments, we used random splitting to divide the downstream datasets into training, validation, and test sets in an 8:1:1 ratio. While random splitting is used consistently across models in this ablation study, scaffold splitting is applied for benchmarking to ensure a fair comparison with other methods.

Table 8: Performance of various models across six MoleculeNet tasks. The data is split using a random split into training, validation, and test sets with an 8:1:1 ratio.

| | BBBP | BACE | HIV | Average | ESOL | FreeSolv | Average | QM9 |
|---|---|---|---|---|---|---|---|---|
| *#tasks* | *1* | *1* | *1* | | *1* | *1* | | *3* |
| *#samples* | *2039* | *1513* | *41127* | | *1128* | *642* | | *133885* |
| *Metric* | | ROC-AUC (↑) | | | | RMSE (↓) | | MAE (↓) |
| **FG_KGE + GAT** | 73.23 ± 1.93 | 76.44 ± 1.27 | 71.65 ± 0.98 | 73.77 | 2.35 ± 0.210 | 4.32 ± 0.29 | 3.335 | 0.0139 ± 0.00014 |
| **AttentiveFP** | 77.71 ± 1.30 | 77.15 ± 0.78 | 78.81 ± 0.99 | 77.89 | 1.63 ± 0.042 | 2.11 ± 0.94 | 1.87 | 0.0056 ± 0.00012 |
| **FG AttentiveFP** | 85.57 ± 1.32 | 87.30 ± 0.90 | 81.21 ± 0.92 | 84.5 | 1.02 ± 0.034 | 1.08 ± 0.14 | 1.05 | 0.0053 ± 0.00034 |
| **BERT** | 82.12 ± 1.45 | 85.12 ± 0.76 | 83.03 ± 1.12 | 83.42 | 1.45 ± 0.056 | 1.89 ± 0.09 | 1.67 | 0.0059 ± 0.00012 |
| **FG BERT** | 94.36 ± 0.50 | 94.54 ± 0.40 | 81.93 ± 1.70 | 90.27 | 0.608 ± 0.031 | 0.507 ± 0.03 | 0.558 | 0.0041 ± 0.00017 |
| **FARM** | 96.23 ± 0.7 | 96.19 ± 0.65 | 82.13 ± 1.10 | **91.43** | 0.734 ± 0.039 | 0.308 ± 0.08 | **0.521** | **0.0038 ± 0.00014** |

Table 9 presents the performance of FARM on the ADMET datasets. The ADMET leaderboard[1] provides a comprehensive evaluation of model performance across ADMET tasks (Absorption - Distribution - Metabolism - Excretion - Toxicity), with a standardized train/validation/test split. For consistency, we use the default train/validation/test split as specified by the ADMET benchmark. FARM achieves state-of-the-art results on 4 out of 16 tasks and demonstrates on-par performance with other top-performing models across the remaining tasks. These results highlight the robustness and competitive nature of FARM in ADMET prediction tasks.

Table 9: FARM's performance on ADMET tasks

| Dataset | ADMET task | Unit | Metric | Task | SOTA | FARM | FARM's ranking |
|---|---|---|---|---|---|---|---|
| Caco2 | | cm/s | MAE | Regression | 0.276 | 0.340 | 10 |
| HIA | | % | AUCROC | Binary | 0.990 | 0.978 | 7 |
| Bioav | Absorption | % | AUROC | Binary | 0.753 | 0.709 | 5 |
| Lipo | | log-ratio | MAE | Regression | 0.467 | 0.523 | 6 |
| **AqSol** | | log mol/L | MAE | Regression | 0.761 | **0.739** | **1** |
| BBB | | % | AUROC | Binary | 0.920 | 0.908 | 7 |
| **PPBR** | Distribution | % | MAE | Regression | 7.526 | **7.376** | **1** |
| VDss | | L/kg | Spearman | Regression | 0.724 | 0.652 | 4 |
| CYP2C9 Inhibition | | % | AUPRC | Binary | 0.859 | 0.798 | 4 |
| CYP3A4 Inhibition | | % | AUPRC | Binary | 0.916 | 0.877 | 5 |
| **CYP2C9 Substrate** | Metabolism | % | AUPRC | Binary | 0.441 | **0.443** | **1** |
| CYP2D6 Substrate | | % | AUPRC | Binary | 0.738 | 0.703 | 5 |
| Half Life | | hr | Spearman | Regression | 0.576 | 0.433 | 8 |
| CL-Hepa | Excretion | uL.min-1.($10^6$ cells)-1 | Spearman | Regression | 0.498 | 0.437 | 8 |
| hERG | | % | AUROC | Binary | 0.880 | 0.793 | 9 |
| **Ames** | Toxicity | % | AUROC | Binary | 0.871 | **0.875** | **1** |

---

[1] https://tdcommons.ai/benchmark/admet_group/overview/

