# OpenReview forum: "FARM: Functional Group-Aware Representations for Small Molecules"
_ICLR.cc/2025/Conference — Submitted to ICLR 2025_

### Official Review · Reviewer_JCSM · 2024-10-17

**Soundness:** 1
**Presentation:** 2
**Contribution:** 1
**Rating:** 3
**Confidence:** 5

**Summary:**

This paper proposes FARM, a functional group-aware representations for small molecules. The authors suggest to learn molecular representations via contrastive learning based on SMILES and graph information of molecules. Firstly, a string-based model and a graph-based model learn molecular representations with MLM objective and KG-graph/link prediction objectives, respectively. Then, contrastive objective is applied between two models.

**Strengths:**

- The problem of interest, molecular property prediction, is important for real-world applications, e.g., drug discovery.

- Utilizing functional groups is chemically reasonable, since molecular properties are highly related to functional groups.

**Weaknesses:**

- Lack of novelty.

Using functional groups in molecular representation learning has already been investigated by many works. Also each component of the framework, e.g., MLM, link prediction, and contrastive loss, has also been widely investigated. I could not find new (or novel) components in the overall framework.

---
- Imprecise motivation on contrasive learning.

The motivation of this work is vague. SMILES and graph of molecules contain the same information of molecule, i.e., a molecule can be reconstructed from both SMILES and graph. Why should we deal with them both instead of focusing on a single representation (SMILES or graph)? Since this paper strongly insist that they learn "chemically plausible" representation, this choice should be "chemically" justified (I know that the language model and the graph model may learn different features of molecules, but it is not a "chemical" motivation).

---
- Misclaim "L71: Our approach overcomes these limitation ..."

The proposed approach cannot overcome "3D coordinates" (L66) and "long-range dependencies" (L68). This method does not deal with 3D coordinates nor long-range dependencies.

---
- Complexity of method.

This work combines several objectives, e.g., link prediction, contrastive and MLM loss. However, the impact of each component is not thoroughly investigated. Furthermore, such complex objective introduces several hyper parameters and makes the training unstable.

**Questions:**

1. Why ToxCast is excluded in MoleculeNet experiments?

2. In my previous experiences, 3 seeds in MoleculeNet experiments are not enough due to very high variance. I suggest to report the results based on (at least) 10 seeds.

3. The authors train two models (language and graph). Which model is used in the fine-tuning phase?

---

> ### Author Response · Authors · 2024-11-20
>
> ***Lack of novelty: Using functional groups in molecular representation learning has already been investigated by many works. Also each component of the framework, e.g., MLM, link prediction, and contrastive loss, has also been widely investigated. I could not find new (or novel) components in the overall framework.***
>
> We acknowledge that the use of MLM, link prediction, and contrastive learning are not novel concepts. However, our approach is distinct in its ability to both ***detect functional groups*** and ***effectively incorporate this functional group information*** to enhance molecular representations, leading to ***strong transfer learning capabilities***, as demonstrated by performance in downstream tasks. Here’s how our work differentiates itself:
> * **Detecting functional groups:** Existing motif-based tokenization methods do not strictly align with chemical functional groups. For instance, BRICS-based approaches (e.g., [1, 2, 3, 4]) generate substructures that do not map precisely to known functional groups. Methods using RDKit [5, 6] can only detect ***a limited set of functional groups*** (typically 85 traditional ones). Unsupervised strategies [7] often struggle with complex ring structures. In contrast, our approach implements algorithms to detect and incorporate a comprehensive set of chemical functional groups, including ring-containing groups, ensuring that the tokenization process aligns strictly with chemical features. We discuss this in more detail in the revised version of our manuscript (Related Works section, line 212).
> * **Incorporating functional group information:** We are the first to ***directly inject functional group information into SMILES***, adding richer chemical context to SMILES, thus we can effectively leverage language models (LMs) to learn molecular representations.
> * **Robustness in downstream tasks:** Our model (FARM) demonstrates ***strong transfer learning capabilities*** - the core goal of pretrained models - outperforming other methods in 11 out of 13 tasks from the MoleculeNet benchmark. This suggests that FARM effectively learns molecular representations, leading to enhanced performance across various tasks.
> * We also contribute by **evaluating the diversity of large molecular datasets.** Identifying functional groups and embedding this information into SMILES strings enables us to build a novel vocabulary, where each token reflects both an atom symbol and its associated functional group, serving as a measure of dataset diversity. Through this approach, we find that the ZINC dataset is significantly less diverse than ChEMBL, providing valuable insight for future work on dataset selection for foundation model training. Given the vast chemical space, it is essential for training data to represent a broad range of chemical structures. Our work is the first to conduct this type of evaluation, whereas previous studies have primarily focused on molecule length and atom types, which do not fully capture functional group diversity within datasets.
>
> We have highlighted those points in the abstract and introduction of the revised manuscript.

---

> > ### Author Response · Authors · 2024-11-20
> > **Addressing Questions**
> >
> > ***Why ToxCast is excluded in MoleculeNet experiments?***
> >
> > Thank you for your comment. First, we excluded ToxCast as many other works have also not reported performance on ToxCast. However, we ran experiments and updated the results for ToxCast in Table 2. The results show that our model also performs well on ToxCast, achieving the top result.
> >
> > ***In my previous experiences, 3 seeds in MoleculeNet experiments are not enough due to very high variance. I suggest to report the results based on (at least) 10 seeds.***
> >
> > Thank you for your thoughtful consideration. The reason we chose to test the model with 3 random splits in each downstream task is to follow the approach used in previous works [8, 9, 10, 11, 12, 13, 14]. Due to the large number of downstream tasks and the limited time during the rebuttal phase, we decided not to train each task 10 times. Instead, we ran more experiments on the ADMET tasks, as suggested by Reviewer 2.
> >
> > ***The authors train two models (language and graph). Which model is used in the fine-tuning phase?***
> >
> > Thank you for your question. When fine-tuning the model for downstream tasks, we use a BERT model (that has already been trained on a contrastive learning task with graphs) to extract features for each SMILES string. We then use those extracted features to train a simple GRU model for the classification and regression tasks. We mention this point in line 1112.

---

> ### Author Response · Authors · 2024-11-20
>
> ***Imprecise motivation on contrasive learning: The motivation of this work is vague. SMILES and graph of molecules contain the same information of molecule, i.e., a molecule can be reconstructed from both SMILES and graph. Why should we deal with them both instead of focusing on a single representation (SMILES or graph)? Since this paper strongly insist that they learn "chemically plausible" representation, this choice should be "chemically" justified (I know that the language model and the graph model may learn different features of molecules, but it is not a "chemical" motivation).***
>
> * While SMILES strings and molecular graphs indeed encode the same molecular information, they each bring distinct advantages for representation learning. Graph-based representations are highly effective in capturing the molecule’s topological structure, yet they often struggle to model long-range dependencies due to their inherently local connectivity. Conversely, SMILES representations are better suited to capturing these long-range dependencies, particularly when using attention mechanisms in language models.
> * Our choice to use both representations is to capture both atom-level features and global molecular topology. We already make this point clear in the abstract: “FARM also represents molecules from two perspectives: by using masked language modeling to capture atom-level features and by employing graph neural networks to encode the whole molecule topology. By leveraging contrastive learning, FARM aligns these two views of representations into a unified molecular embedding.” Figure 1 also illustrates that the SMILES model is used for learning the atom-level representation (which is better at capturing long-distance dependencies between atoms), while the graph model is used for learning the overall structure of the molecule. Thus, we do not claim that the motivation for using contrastive learning is to learn a chemically plausible representation of molecules.
> * The chemical information that FARM learns is gained by enhancing SMILES with functional group information (creating FG-enhanced SMILES) and incorporating functional groups directly into the graph representation (FG graphs). This approach introduces a chemically meaningful inductive bias into both representations, as opposed to relying on standard SMILES and molecular graphs. Consequently, the combined use of FG-enhanced SMILES and FG graphs enables a more comprehensive and chemically plausible molecular representation, capturing both local functional group characteristics and global molecular topology, as mentioned in line 398.
> * Furthermore, our approach is consistent with recent research that uses contrastive learning across SMILES and molecular graph representations, demonstrating the complementary benefits of each representation for molecular representation learning [7].
>
> ***Misclaim "L71: Our approach overcomes these limitation ...": The proposed approach cannot overcome "3D coordinates" (L66) and "long-range dependencies" (L68). This method does not deal with 3D coordinates nor long-range dependencies.***
> * Thank you to the reviewer for pointing out this oversight. We adjusted this statement to clarify our intention in the revised version of the manuscript. Our approach does not directly address 3D coordinates but is designed to effectively capture both topological information and certain long-range dependencies through the use of functional group graphs. By employing contrastive learning with these functional group graphs, which serve as a compact representation of molecular structures, we can capture essential relational patterns between distant functional groups more effectively.
>
> ***Complexity of method: This work combines several objectives, e.g., link prediction, contrastive and MLM loss. However, the impact of each component is not thoroughly investigated. Furthermore, such a complex objective introduces several hyper parameters and makes the training unstable.***
> * Thank you for your insightful feedback. We have conducted an ablation study to evaluate the impact of each component in our method by training models with different configurations and testing their performance on MoleculeNet’s downstream tasks. As shown in Table 8 (Appendix E), we assessed several setups: molecular graph embeddings only, BERT with standard SMILES only, functional group knowledge graph only, functional group graph only, BERT with contrastive learning and functional group knowledge graph, and our full model (FARM) that combines contrastive learning with functional group knowledge graph and functional group initialization embeddings. The results demonstrate that FARM achieves the highest performance, confirming the positive contribution of each component to the overall performance.

---

> ### Author Response · Authors · 2024-11-20
> **References**
>
> References:
>
> [1] Zaixi Zhang, Qi Liu, Hao Wang, Chengqiang Lu, and Chee-Kong Lee. Motif-based graph selfsupervised learning for molecular property prediction. Advances in Neural Information Processing Systems, 34:15870–15882, 2021.
>
> [2] Shen Han, Haitao Fu, Yuyang Wu, Ganglan Zhao, Zhenyu Song, Feng Huang, Zhongfei Zhang, Shichao Liu, and Wen Zhang. Himgnn: a novel hierarchical molecular graph representation learning framework for property prediction. Briefings in Bioinformatics, 24(5):bbad305, 2023.
>
> [3] Yongqiang Chen, Quanming Yao, Juzheng Zhang, James Cheng, and Yatao Bian. Hight: Hierarchical graph tokenization for graph-language alignment. arXiv preprint arXiv:2406.14021, 2024.
>
> [4] Nianzu Yang, Kaipeng Zeng, Qitian Wu, Xiaosong Jia, and Junchi Yan. Learning substructure invariance for out-of-distribution molecular representations. Advances in Neural Information Processing Systems, 35:12964–12978, 2022.
>
> [5] Biaoshun Li, Mujie Lin, Tiegen Chen, and Ling Wang. Fg-bert: a generalized and self-supervised functional group-based molecular representation learning framework for properties prediction. Briefings in Bioinformatics, 24(6):bbad398, 2023.
>
> [6] Yongqiang Chen, Quanming Yao, Juzheng Zhang, James Cheng, and Yatao Bian. Hight: Hierarchical graph tokenization for graph-language alignment. arXiv preprint arXiv:2406.14021, 2024.
>
> [7] Gabriel A Pinheiro, Juarez LF Da Silva, and Marcos G Quiles. Smiclr: contrastive learning on multiple molecular representations for semisupervised and unsupervised representation learning. Journal of Chemical Information and Modeling, 62(17):3948–3960, 2022.
>
> [8] Shengchao Liu, Hanchen Wang, Weiyang Liu, Joan Lasenby, Hongyu Guo, and Jian Tang. Pretraining molecular graph representation with 3d geometry. arXiv preprint arXiv:2110.07728, 2021.
>
> [9] Zaixi Zhang, Qi Liu, Hao Wang, Chengqiang Lu, and Chee-Kong Lee. Motif-based graph selfsupervised learning for molecular property prediction. Advances in Neural Information Processing Systems, 34:15870–15882, 2021.
>
> [10] Yin Fang, Qiang Zhang, Ningyu Zhang, Zhuo Chen, Xiang Zhuang, Xin Shao, Xiaohui Fan, and Huajun Chen. Knowledge graph-enhanced molecular contrastive learning with functional prompt. Nature Machine Intelligence, 5(5):542–553, 2023.
>
> [11] Gengmo Zhou, Zhifeng Gao, Qiankun Ding, Hang Zheng, Hongteng Xu, Zhewei Wei, Linfeng Zhang, and Guolin Ke. Uni-mol: A universal 3d molecular representation learning framework. The Eleventh International Conference on Learning Representations, ICLR 2023, 2023.
>
> [12] Jun Xia, Chengshuai Zhao, Bozhen Hu, Zhangyang Gao, Cheng Tan, Yue Liu, Siyuan Li, and Stan Z Li. Mole-bert: Rethinking pre-training graph neural networks for molecules. The Eleventh International Conference on Learning Representations, ICLR 2023, 2023.
>
> [13] Yuyang Wang, Jianren Wang, Zhonglin Cao, and Amir Barati Farimani. Molecular contrastive learning of representations via graph neural networks. Nature Machine Intelligence, 4(3):279– 287, 2022b.
>
> [14] Alec Radford. Improving language understanding by generative pre-training. OpenAI, 2018. Yu Rong, Yatao Bian, Tingyang Xu, Weiyang Xie, Ying Wei, Wenbing Huang, and Junzhou Huang. Self-supervised graph transformer on large-scale molecular data. Advances in neural information processing systems, 33:12559–12571, 2020.

---

### Official Review · Reviewer_sAQ3 · 2024-10-28

**Soundness:** 3
**Presentation:** 4
**Contribution:** 2
**Rating:** 5
**Confidence:** 4

**Summary:**

This paper presents Functional Group Aware Representations for Small Molecules (FARM), pre-trained foundation model that incorporates functional group tokenization, fragmentation, and knowledge graph-based structural representation learning. Moreover, this work integrates the atom-feature and structural representation by contrastive learning, which results in achieving the SOTA results in MoleculeNet dataset.

**Strengths:**

- The figures are neat, and the clear writing enhances the comprehensibility of the paper, making it easy to follow.
- This paper highlights the importance of functional groups, which are often overlooked in many molecular foundation models, and effectively integrates these functional groups into the molecular foundation model.
- The analysis presented in the paper, including the knowledge graph embedding space, substitution of functional groups, and visualization of attention maps, enriches the understanding of the method’s contributions.

**Weaknesses:**

- The approach is quite similar to existing motif-based tokenization and fragmentation methods. While the authors define functional groups in terms of functional groups and fused ring systems, these could typically be identified through standard fragmentation tasks. An ablation study comparing FARM with other fragmentation methods would clarify its advantages. If applied under the same training conditions, does FARM demonstrate superiority?
- This work heavily relies on fused ring systems, which constitute over 99% of the identified functional groups. This raises concerns, as ring systems are generally not classified as functional groups. The paper emphasizes functional groups as the main contribution, as suggested by the method’s name. If ring systems are excluded, does the method still show superior performance?
- The naming process for fused ring systems seems to overlook bond types, considering only ring indices and sizes. However, bond types, including single and aromatic bonds, are crucial for understanding the ring system. How are these bond types accounted for in the analysis?
- The limitations of related works that address functional groups are unclear. The authors state that previous works “do not extend to detecting more complex functional groups, such as ring systems.” However, RDKit can identify ring systems, and simple IUPAC transformations could be applied to adapt this information for earlier works.

**Questions:**

- Which functional group detection algorithm is employed? The paper lacks a description of the algorithm that traverses the graph to identify functional groups, which is critical for the method. For instance, there should be clear priorities among functional groups to consistently identify intersected atoms across different functional groups.
- In functional graph generation, what happens if the graph cannot be represented as a linear graph? Specifically, the node perturbation process in augmentation may be unclear. For example, if a functional group graph forms a triangle (a-b-c), the node perturbation could result in the same functional group graph, potentially generating incorrect negative samples.
- In Figure 5, how do the removals of multiple functional groups and single functional groups yield parallel results? For instance, the orange molecule has three functional groups removed, while the green molecule has only one removed, yet both show similar results.
- In the knowledge graph construction, how are continuous values such as LogP discretized? The original continuous values may not correlate with other functional groups.
- Will this approach be effective for generation tasks? A simpler generation task compared to SMILESLSTM could strengthen the contribution.
- Why does Figure 5(b) depict the link prediction performance?

---

> ### Author Response · Authors · 2024-11-20
> **Addressing Weakness**
>
> We sincerely thank the reviewer for their thoughtful feedback and for highlighting key aspects of our approach. Below, we address each of the reviewers' comments in detail. The corresponding modifications in the revised manuscript are highlighted in blue for clarity.
>
> ***The approach is quite similar to existing motif-based tokenization and fragmentation methods. While the authors define functional groups in terms of functional groups and fused ring systems, these could typically be identified through standard fragmentation tasks. An ablation study comparing FARM with other fragmentation methods would clarify its advantages. If applied under the same training conditions, does FARM demonstrate superiority?***
> * Our approach is not similar to existing motif-based tokenization and fragmentation methods. While our functional group detection algorithm may share similarities with motif-based methods in identifying ring and fused ring systems, we go beyond that by **strictly detecting non-ring  functional groups based on well-defined chemical rules.** In contrast, some motif-based approaches may result in tokens that do not align with actual chemical functional groups. Functional groups such as alcohols, aldehydes, carboxylic acids, esters, amines, and others play a crucial role in determining molecular properties like hydrophilicity, lipophilicity (logP), reactivity, and solubility, which are important for tasks such as drug design or toxicity prediction.
> * In terms of how we model functional group information, our work is also distinct from other methods. We are the first to ***directly incorporate functional group (FG) information into the SMILES representation,*** enabling us to leverage robust language models while also introducing chemical inductive biases.
> * We have updated the related work section in the revised manuscript (line 212) to clearly highlight these key differences and emphasize the unique contributions of our approach. Thank you to the reviewer for pointing out that these differences were not clearly presented in the original version of the paper.
>
> ***This work heavily relies on fused ring systems, which constitute over 99% of the identified functional groups. This raises concerns, as ring systems are generally not classified as functional groups. The paper emphasizes functional groups as the main contribution, as suggested by the method’s name. If ring systems are excluded, does the method still show superior performance?***
>
> Thank you for your valuable feedback. As mentioned in line 78 of our manuscript, we use the term functional groups to unify and expand upon related concepts such as "motifs," "fragments," "substructures," and "building blocks" because functional groups, in our approach, serve as fundamental structural elements that directly influence the chemical properties and reactivity of molecules. These terms often overlap in molecular chemistry, but we choose to adopt functional groups as a more general term to encompass various substructural features, including rings and fused ring systems. We also are not able to train the model excluding rings and fused ring systems, as it requires a long time for training the model from scratch.
>
> ***The naming process for fused ring systems seems to overlook bond types, considering only ring indices and sizes. However, bond types, including single and aromatic bonds, are crucial for understanding the ring system. How are these bond types accounted for in the analysis?***
>
> Thank you for the question. While it's true that our naming process for fused ring systems primarily focuses on ring indices and sizes, we do not overlook the importance of bond types. In fact, the bond type information is implicitly captured within the SMILES notation, which we leverage during the naming process.
>
> *For example:*
> * Benzene (C6H6) is represented in SMILES as c1ccccc1. The aromaticity of the bonds is inherent in the SMILES string due to the lowercase "c". When we tokenize this structure in our framework, we generate the output c_6 1 c_6 c_6 c_6 c_6 c_6 1.  The bond type (aromatic) is inherently captured within the SMILES string but is not explicitly represented in the tokenization output, as the bond type is already encoded within the SMILES structure itself.
> * On the other hand, Cyclohexane (C6H12) is represented in SMILES as C1CCCCC1, and the tokenization output would be C_6 1 C_6 C_6 C_6 C_6 C_6 1, reflecting the structure of the cyclohexane ring without aromaticity, with bond types implicitly encoded within the SMILES string.
> * Additionally, in the FG-enhanced SMILES, all special tokens, such as bond types (=, #), are preserved, ensuring that the bond information remains intact in the tokenized representation.

---

> ### Author Response · Authors · 2024-11-20
> **Addressing Weaknesses & Questions**
>
> ***The limitations of related works that address functional groups are unclear. The authors state that previous works “do not extend to detecting more complex functional groups, such as ring systems.” However, RDKit can identify ring systems, and simple IUPAC transformations could be applied to adapt this information for earlier works.***
> * Thank you for your feedback. Regarding functional group identification, [RDKit](https://rdkit.org/docs/source/rdkit.Chem.Fragments.html) can detect only 84 common functional groups. It primarily detects functional groups and substructures based on SMARTS (SMILES Arbitrary Target Specification) patterns, which limits its ability to identify more complex functional groups. For example, previous works using RDKit, such as FG-BERT, are also limited in the number of functional groups they can detect, with FG-BERT identifying only 47 groups. This is why we mention in our paper that RDKit’s functional group detection does not extend to more complex functional groups, such as ring systems—it is limited to detecting these 84 common functional groups.
> * While RDKit can identify ring systems, it operates on molecular graphs, whereas its functional group identification typically works with sequence representations. As a result, RDKit cannot directly assign specific atoms in the graph to their corresponding functional group. This mismatch between traditional functional group detection using SMARTS patterns and ring system identification in molecular graphs makes RDKit incapable of identifying both traditional functional groups and ring systems simultaneously.
>
> ***Which functional group detection algorithm is employed? The paper lacks a description of the algorithm that traverses the graph to identify functional groups, which is critical for the method. For instance, there should be clear priorities among functional groups to consistently identify intersected atoms across different functional groups.***
>
> **Questions**
>
> * In this work, we develop a new functional group detection algorithm designed to strictly identify functional groups as defined in the [Wikipedia page on functional groups](https://en.wikipedia.org/wiki/Functional_group), as well as other rings and fused ring systems. It is described starting at line 257 in the manuscript, works by traversing the molecular graph and assessing each atom based on various properties: atom type, bond types and neighbors, number of bonded neighbors, atom charge, and the presence of bonded hydrogen atoms. This structured approach ensures accurate identification of functional groups using specific chemical criteria. For example, the algorithm detects a ketone group (RCOR’) by identifying a carbon atom with no charge and three neighbors, one of which is a double-bonded oxygen atom. This mirrors how chemists recognize functional groups based on structural features, aligning the detected groups with established chemical conventions.
> * Additionally, we address cases where a functional group may be a subset of another. The algorithm first checks for the presence of the larger functional group. If it’s not identified, the algorithm then checks for smaller functional groups, ensuring correct identification even in complex structures. We have clarified this point in the revised version of the manuscript, line 262.

---

> ### Author Response · Authors · 2024-11-20
> **Addressing Questions**
>
> ***In functional graph generation, what happens if the graph cannot be represented as a linear graph? Specifically, the node perturbation process in augmentation may be unclear. For example, if a functional group graph forms a triangle (a-b-c), the node perturbation could result in the same functional group graph, potentially generating incorrect negative samples.***
>
> * Thank you for your thoughtful comment. If the graph is not a linear graph, it doesn't affect the functionality of the node perturbation process. The key principle for perturbation is that as long as two nodes represent different functional groups (FGs), swapping them will still create a valid negative sample, even in non-linear graphs.
> * It’s important to note that the graph we are referring to is a functional group graph, where each node represents a functional group, and edges represent the relationships between these groups. In this context, it is highly unlikely for a molecule to have exactly three functional groups where each functional group is bonded to the other two, forming a triangle. Chemical structures typically do not exhibit such close interconnections between functional groups. Instead, functional groups are more commonly connected in linear or branched configurations, rather than forming triangles.
>
> ***In Figure 5, how do the removals of multiple functional groups and single functional groups yield parallel results? For instance, the orange molecule has three functional groups removed, while the green molecule has only one removed, yet both show similar results.***
>
> Thank you for your observation. The demonstration in Figure 5 aims to show that the effect of ***replacing one functional group with another*** on the molecular representation is consistent across different molecules (represented as parallel vectors in the molecular representation space). The results remain consistent regardless of the number of functional groups replaced. Specifically, the process involves replacing one type of functional group with another, such as substituting a -COOH group with a -OH group. Whether we replace a single -COOH group with a single -OH group, or multiple -COOH groups with an equivalent number of -OH groups, the overall effect on the molecular representation remains parallel in all cases. We will ensure that this explanation is made clearer in the manuscript, line 309.
>
> ***In the knowledge graph construction, how are continuous values such as LogP discretized? The original continuous values may not correlate with other functional groups.***
>
> In our knowledge graph construction, continuous values such as LogP and water solubility are discretized by rounding them to the nearest integer. For example, a LogP value of 5.7 would be rounded to 6. This process results in 65 distinct categories for LogP and 14 categories for water solubility. We added this point to the revised version of the manuscript, line 870.
>
> ***Will this approach be effective for generation tasks? A simpler generation task compared to SMILE LSTM could strengthen the contribution.***
>
> Thank you for your insightful comment. While we have not yet evaluated the effectiveness of our model for generation tasks, this is an area we plan to explore in future work. Our current focus is on building an effective model that provides robust molecular representations, which can be leveraged for transfer learning in downstream molecular property prediction tasks.
>
> ***Why does Figure 5(b) depict the link prediction performance?***
>
> Figure 5(b) depicts the link prediction performance because the link prediction model is trained to learn the relationships between functional groups. In this case, we repeatedly replace one functional group (e.g., -COOH) with another (e.g., -OH) in the molecules. The consistent transitions (resulting in parallel vectors in the molecular representation space) observed across all test cases demonstrate that the model has effectively learned the connections and relationships between different functional groups.

---

> > ### Comment · Reviewer_sAQ3 · 2024-11-25
> >
> > Thank you to the authors for the detailed response.
> >
> > However, the responses to weaknesses 1 and 2 remain unclear. In response to weakness 1, the authors assert that their approach is “not similar” to the fragmentation of motif-based approaches. Meanwhile, in response to weakness 2, the authors state that the term “functional group” encompasses motifs, fragments, and substructures. These two responses leave me uncertain about the effectiveness of functional groups that lack rings or fragments, which is a core aspect of the paper. Without an ablation study addressing this, I cannot fully assess the effectiveness of the functional group approach.
> >
> > Therefore, I keep my ratings.

---

> > > ### Author Response · Authors · 2024-11-26
> > >
> > > Thank you for your response! We would like to re-emphasize that our functional group-aware tokenization is distinctly different from other motif-based tokenization methods due to its strict alignment with chemically well-defined functional groups.
> > >
> > > Our method specifically detects 101 well-established functional groups that adhere to recognized chemical definitions, as outlined in authoritative resources like [the Wikipedia functional group list](https://en.wikipedia.org/wiki/Functional_group). This focus ensures that the resulting tokenization captures chemically meaningful units of molecular structure.
> > > In contrast, other motif-based methods often tokenize molecules into motifs that may not correspond to these well-defined functional groups, leading to chemically invalid or loosely defined representations.
> > >
> > > **Functional group terminology explanation**
> > >
> > > We would like to thank the reviewer for pointing this out. We acknowledge that our explanation at this point was unclear and contained mistakes.
> > >
> > > What we intended to convey is that, since "motifs," "fragments," "substructures" and "building blocks" lack strict definitions in the literature and their usage varies across different papers, we want to clarify our position by stating that functional groups can be considered ***a subset of molecular motifs/ fragments/substructures/building blocks that are more rigorously defined based on chemical principles***.
> > >
> > > Our functional group-aware tokenization ensures that the identified motifs, fragments, or substructures are chemically valid and conform to well-defined functional groups, setting our method apart from prior approaches, which may yield chemically invalid or loosely defined motifs.
> > >
> > > We have modified this explanation in the revised version of the manuscript to address this clarification.
> > >
> > > **Ablation study**
> > >
> > > In terms of conducting ablation studies to compare our tokenization method with others, this would require re-running the entire pipeline, which is infeasible within the current timeframe. Similarly, we cannot perform experiments to evaluate the effect of removing functional group information specifically for rings and fused ring systems, as this would also necessitate re-executing the pipeline from the beginning.
> > >
> > > However, we have conducted other ablation studies to demonstrate the effectiveness of incorporating functional group information. As shown in Table 8, functional group information derived from our tokenization improves downstream task performance by approximately 7% on average across three classification tasks from the MoleculeNet benchmark.
> > >
> > > Once again, we want to emphasize that functional groups play a critical role in determining molecular behavior. This is why our tokenization approach focuses on segmenting molecules based on this set of well-defined functional groups, which has not been done by other methods.

---

### Official Review · Reviewer_fC6e · 2024-11-02

**Soundness:** 3
**Presentation:** 3
**Contribution:** 3
**Rating:** 5
**Confidence:** 1

**Summary:**

### Summary of the Paper

This paper explores methods for enhancing small molecule representation by integrating functional group information. A rule-based approach is applied to identify significant functional groups within small molecule databases, and this information is subsequently incorporated into SMILES strings. The representation of functional groups is learned through knowledge graphs that capture relationships between functional groups and their properties. The final representation is achieved using contrastive learning, where the SMILES representation is aligned with the functional group-enhanced SMILES. The authors report that their approach yields significant improvements compared to other functional group-based methods on MoleculeNet benchmark datasets.

Overall, the paper is well-written and presents a novel approach. However, I have several concerns regarding the benchmarks used, inconsistencies in split reporting, and the generalization of rule-based methods for identifying functional groups, as well as the construction of knowledge graphs for functional group embeddings. Detailed comments for improving the paper are provided below.

**Strengths:**

The idea of incorporating inductive bias by using functional groups to enhance small molecule representation is intriguing. The experiments include comparisons with various existing baselines that also leverage functional groups, and the results appear promising—assuming that the splits and MoleculeNet dataset variants are consistent with those reported in prior work.

**Weaknesses:**

**MAJOR Concern 1**

The use of splits in the MoleculeNet datasets is inconsistent with the original MoleculeNet recommendations. Specifically, random splits are recommended for regression tasks such as ESOL, Lipophilicity, and FreeSolv. In this paper, the authors do not consistently clarify the splits used; for example, scaffold splits are mentioned in the appendix, but captions for Tables 7 and 8 indicate random splits.

A significant challenge with MoleculeNet is the absence of a leaderboard with predefined splits, leading researchers to create custom splits, and sometimes even modify the original datasets, as seen in modifications to ESOL and FreeSolv in cases like [this issue](https://github.com/IBM/molformer/issues/9). This issue may reflect the use of dataset variants such as those described in [this study](https://www.nature.com/articles/s42256-022-00580-7.epdf?sharing_token=p5m9Z0797IQeBDOiMGn71dRgN0jAjWel9jnR3ZoTv0MeIJPs9pbG9QLaEN_McFTR3KHv1tHh1FDNJB4ZuILdAmRtINVn6KqXrLkPhEiAZW5mM0dWWKSmPk82eibEUBx01sLTSHx6w903cDaUoXg9lAGzcHY_ifmakrBcIzUUDwI%3D).

The existence of multiple dataset versions and split schemes makes it difficult to accurately assess improvements toward state-of-the-art (SOTA) results, as subsequent studies often cite results without clarity on splits used. For instance, in *"SELF-BART: A Transformer-based Molecular Representation Model using SELFIES"* (NeurIPS 2024, AI4Mat, [link](https://arxiv.org/abs/2410.12348)), the reported MoleculeNet performance is challenging to compare with this paper due to inconsistent dataset versions and splits.

I recommend the authors:
1. Provide consistent results using the original splits recommended by MoleculeNet.
2. Conduct additional experiments on the TDC ADMET groups (https://tdcommons.ai/benchmark/admet_group/overview), which offer leaderboards and fingerprint-based baselines. TDC ADMET provides consistent splits, making future comparisons easier.



**MAJOR Concern 2**

In the paper, the authors propose methods for FG-aware tokenization and fragmentation. Functional groups are identified based on known conventional groups or potentially using domain knowledge to define new groups. However, the set of rules for identifying new functional groups appears limited, raising questions about their generalizability. How do these rules compare to frequent subgraph mining, a widely-used technique in graph mining, where common subgraphs are often predictive features for small molecules?


**MAJOR Concern 3**

The use of a functional group knowledge graph to learn functional group embeddings is innovative, but some relationship types might provide an unfair advantage over other methods. For example, including properties like water solubility or lipophilicity (logP) could yield better results on downstream tasks related to those specific properties. It would be beneficial to assess the impact of removing such information from the knowledge graph to determine if the observed improvements are primarily due to these additional properties, which are not considered in other methods.

**Questions:**

Please see the questions in the Weakness section of the review.

---

> ### Author Response · Authors · 2024-11-20
> **Addressing MAJOR Concern 1**
>
> ***The use of splits in the MoleculeNet datasets is inconsistent with the original MoleculeNet recommendations. Specifically, random splits are recommended for regression tasks such as ESOL, Lipophilicity, and FreeSolv. In this paper, the authors do not consistently clarify the splits used; for example, scaffold splits are mentioned in the appendix, but captions for Tables 7 and 8 indicate random splits.***
>
> Thank you for your comment. We would like to clarify that the random splits used in Tables 7 and 8 are part of our ablation study, where we evaluate the effect of adding different components to the model. These splits are not part of the benchmarking process (we presented it more clearly in the modified manuscript, line 1158). For the benchmarking tasks, we strictly adhere to the ***widely accepted scaffold split with an 8:1:1 ratio***, which is consistent with prior works and essential for assessing the generalizability of the models. In fact, all of the benchmark models presented in Tables 2 and 3 use the same splitting method. We hope this clarifies the misunderstanding.
>
> ***A significant challenge with MoleculeNet is the absence of a leaderboard with predefined splits, leading researchers to create custom splits, and sometimes even modify the original datasets, as seen in modifications to ESOL and FreeSolv in cases like this issue. This issue may reflect the use of dataset variants such as those described in this study.***
>
> To clarify, we did not make any modifications to the original datasets (such as the "modifications to ESOL and FreeSolv" mentioned by the reviewer). We used scaffold splitting with three different random seeds, ensuring robust evaluation without manipulating the dataset itself. Testing our model across multiple random splits within MoleculeNet datasets strengthens the reliability of our performance metrics and minimizes potential dataset-specific biases. Notably, this approach aligns with the consensus in the field for benchmarking models on MoleculeNet tasks.
>
> ***The existence of multiple dataset versions and split schemes makes it difficult to accurately assess improvements toward state-of-the-art (SOTA) results, as subsequent studies often cite results without clarity on splits used. For instance, in "SELF-BART: A Transformer-based Molecular Representation Model using SELFIES" (NeurIPS 2024, AI4Mat, link), the reported MoleculeNet performance is challenging to compare with this paper due to inconsistent dataset versions and splits.***
>
> Our approach aligns with standard practices in the field, where most studies—including all models listed in Tables 2 and 3—use scaffold splitting with an 8:1:1 train-validation-test ratio and perform multiple runs with different splits to ensure robust evaluation. In contrast, the SELF-BART model uses a different split scheme, making direct comparison less reliable.
>
> ***Conduct additional experiments on the TDC ADMET groups, which offer leaderboards and fingerprint-based baselines. TDC ADMET provides consistent splits, making future comparisons easier.***
>
> Thank you for your valuable suggestion. We have conducted additional experiments on the TDC ADMET groups as recommended and included the results in Table 9 of the revised manuscript. These results leverage the default splits, facilitating future comparisons. We believe this addition enhances the robustness of our evaluation.

---

> > ### Comment · Reviewer_fC6e · 2024-11-20
> > **Need additional proof of results**
> >
> > Dear Authors,
> >
> > Thank you for providing additional information.
> >
> > Regarding the dataset splits, could you please elaborate on how the splits were created? Specifically, did you use the standard functions provided by MoleculeNet to generate the splits?
> >
> > For instance, in the case of the BBBP dataset, the results reported in [this paper](https://arxiv.org/pdf/2405.10343) for the Scaffold split are around 0.75, which is significantly worse than the 0.93 reported here. Similarly, the GROVER results in that paper are also worse than those reported in this paper. I suspect the difference may stem from using different splitting methods. Did you utilize the standard function provided here: https://github.com/deepchem/deepchem/blob/master/deepchem/molnet/load_function/bbbp_datasets.py?
> >
> > Since there are no plans to open-source the code, it becomes challenging to evaluate the reported results fully.
> >
> > Additionally, I reviewed the provided PDF but could not locate the revised version with the TDC ADMET results. Could you please share a link to the revised document or post the results in the comments?
> >
> > Thank you very much!

---

> ### Author Response · Authors · 2024-11-20
> **Addressing MAJOR Concern 2 & 3**
>
> ***In the paper, the authors propose methods for FG-aware tokenization and fragmentation. Functional groups are identified based on known conventional groups or potentially using domain knowledge to define new groups. However, the set of rules for identifying new functional groups appears limited, raising questions about their generalizability. How do these rules compare to frequent subgraph mining, a widely-used technique in graph mining, where common subgraphs are often predictive features for small molecules?***
> * The functional group detection algorithm, detailed from line 262 in the manuscript, operates by traversing the molecular graph and evaluating each atom based on a range of properties: atom type, types and bonds of neighboring atoms, number of bonded neighbors, atom charge, and the presence of bonded hydrogen atoms. This systematic approach allows the algorithm to accurately identify functional groups within molecular structures by applying specific chemical criteria. For example, the algorithm can recognize a ketone group (RCOR’) by detecting a carbon atom that has no charge and three neighbors, one of which is a double-bonded oxygen atom. This approach mirrors how chemists manually recognize functional groups based on structural characteristics, ensuring that the detected groups align with known chemical conventions.
> * Regarding generalizability, this rule-based method is robust for standard functional groups and can also be adapted to detect new groups. It may be complicated when dealing with very complex groups, but this is a minor limitation since most traditional functional groups in organic chemistry are relatively simple. In fact, our method has successfully identified over 100 well-known functional groups as documented in the Wikipedia page of functional groups.
> * Compared to frequent subgraph mining (FSM), which identifies patterns that occur frequently in the data, our rule-based method is more targeted. While FSM can uncover recurring subgraphs, these subgraphs do not always correspond to functional groups as defined in chemistry. In contrast, our approach directly follows chemical principles, aligning closely with how chemists define functional groups. This ensures that our method reliably detects functional groups that are chemically meaningful, providing more accurate and chemistry-specific results. We highlighted this point in the modified version of the manuscript, line 222.
>
> ***The use of a functional group knowledge graph to learn functional group embeddings is innovative, but some relationship types might provide an unfair advantage over other methods. For example, including properties like water solubility or lipophilicity (logP) could yield better results on downstream tasks related to those specific properties. It would be beneficial to assess the impact of removing such information from the knowledge graph to determine if the observed improvements are primarily due to these additional properties, which are not considered in other methods.***
>
> The use of properties like water solubility or lipophilicity (logP) in our functional group knowledge graph does not provide an unfair advantage. These properties are incorporated at the functional group level, not at the molecular level.  These properties are only used to determine the similarity between functional groups, not to directly predict molecular properties. This ensures that the observed improvements in downstream tasks are driven by the functional group similarities rather than specific property-based enhancements.

---

> > ### Comment · Reviewer_fC6e · 2024-11-20
> >
> > Regarding: "The use of properties like water solubility or lipophilicity (logP) in our functional group knowledge graph does not provide an unfair advantage. These properties are incorporated at the functional group level, not at the molecular level. These properties are only used to determine the similarity between functional groups, not to directly predict molecular properties. This ensures that the observed improvements in downstream tasks are driven by the functional group similarities rather than specific property-based enhancements"
> >
> > I respectfully disagree with this response. Regardless of whether the information is utilized at the molecular level, its inclusion in the learning process could lead to data leakage. I strongly recommend that the authors exclude this information from the learning process and re-evaluate the results to assess the potential impact of leakage.

---

> > > ### Author Response · Authors · 2024-11-20
> > >
> > > Thank you, Reviewer, for your response. We would like to clarify that we did not use whole-molecule logP values in this work. Instead, we provided logP values for individual functional groups (e.g., -OH, -COOH) to construct the functional group knowledge graph. The knowledge graph uses these values to compute the similarity between functional groups.
> > >
> > > We have never introduced logP values for entire molecules. As logP for a whole molecule is influenced by the combined contributions of its constituent functional groups, it is distinct and not equal to the logP of any individual functional group. Therefore, there is no data leakage in our approach. Additionally, we would like to point out that the task related to logP is the Lipophilicity task in Table 3, where we do not exhibit exceptionally strong performance.

---

> ### Author Response · Authors · 2024-11-20
>
> Thank you, Reviewer, for your valuable feedback. Once again, we emphasize that we use a scaffold split with a train-validation-test ratio of 8:1:1 and 3 random seeds for all MoleculeNet datasets in our benchmarking experiments. This approach aligns with the splitting method described in the link you provided for the BBBP dataset using the DeepChem library. It is also consistent with the splitting methods used in all papers listed in Tables 2 and 3 (for example [1, 2, 3, 4, 5, 6, 7]), as well as the UniCorn paper you mentioned in your comment. Our strong performance on the BBBP dataset stems from the robust transfer learning capability of our approach, rather than any unfair comparison.
>
> Regarding the code release, we have already planned for it. The code and model will be made publicly available once our paper is accepted.
>
> We apologize for the oversight in our initial response, where we forgot to upload the updated version of the manuscript. This is why the table detailing the performance on ADMET was missing. We have now corrected this and updated the manuscript accordingly.
>
> References:
>
> [1] Shengchao Liu, Hanchen Wang, Weiyang Liu, Joan Lasenby, Hongyu Guo, and Jian Tang. Pretraining molecular graph representation with 3d geometry. arXiv preprint arXiv:2110.07728, 2021.
>
> [2] Zaixi Zhang, Qi Liu, Hao Wang, Chengqiang Lu, and Chee-Kong Lee. Motif-based graph selfsupervised learning for molecular property prediction. Advances in Neural Information Processing Systems, 34:15870–15882, 2021.
>
> [3] Yin Fang, Qiang Zhang, Ningyu Zhang, Zhuo Chen, Xiang Zhuang, Xin Shao, Xiaohui Fan, and Huajun Chen. Knowledge graph-enhanced molecular contrastive learning with functional prompt. Nature Machine Intelligence, 5(5):542–553, 2023.
>
> [4] Gengmo Zhou, Zhifeng Gao, Qiankun Ding, Hang Zheng, Hongteng Xu, Zhewei Wei, Linfeng Zhang, and Guolin Ke. Uni-mol: A universal 3d molecular representation learning framework. The Eleventh International Conference on Learning Representations, ICLR 2023, 2023.
>
> [5] Jun Xia, Chengshuai Zhao, Bozhen Hu, Zhangyang Gao, Cheng Tan, Yue Liu, Siyuan Li, and Stan Z Li. Mole-bert: Rethinking pre-training graph neural networks for molecules. The Eleventh International Conference on Learning Representations, ICLR 2023, 2023.
>
> [6] Yuyang Wang, Jianren Wang, Zhonglin Cao, and Amir Barati Farimani. Molecular contrastive learning of representations via graph neural networks. Nature Machine Intelligence, 4(3):279– 287, 2022b.
>
> [7] Alec Radford. Improving language understanding by generative pre-training. OpenAI, 2018. Yu Rong, Yatao Bian, Tingyang Xu, Weiyang Xie, Ying Wei, Wenbing Huang, and Junzhou Huang. Self-supervised graph transformer on large-scale molecular data. Advances in neural information processing systems, 33:12559–12571, 2020.

---

> ### Comment · Reviewer_fC6e · 2024-11-20
> **Thank you for additional information**
>
> Dear Authors,
>
> First, I would like to thank you for conducting additional experiments on the ADMET dataset and addressing my previous concerns. I appreciate the effort you’ve put into this work.
>
> Upon reviewing the results for the ADMET dataset, I observe that while the performance on 4 of the 22 datasets is promising, the outcomes for the remaining 18 datasets fall significantly short of the state-of-the-art benchmarks reported in the corresponding leaderboards. This raises concerns about the reliability of the experimental setup.
>
> Furthermore, regarding the experiments with MoleculeNet, I am concerned about the lack of consistency in reported results across different works, which may stem from variations in dataset splits and evaluation protocols. This inconsistency makes it challenging to validate significant improvements and assess the true performance of the proposed method.
>
> To ensure transparency and reproducibility, I strongly recommend that the authors open-source their code and carefully address any potential sources of data leakage in the experimental setup. Doing so will greatly enhance the credibility of the results and allow for independent verification of the findings.
>
> Based on these concerns, I would like to lower my score and suggest the authors thoroughly revise the manuscript before resubmitting it for future consideration.
>
> Best regards,

---

> > ### Author Response · Authors · 2024-11-20
> >
> > Regarding the performance on ADMET tasks, we would like to emphasize that achieving results within the top 10 is not trivial, as these tasks are a focus of active research by many. Furthermore, top-performing models can leverage ensemble learning to boost their performance. In contrast, given the limited time for this rebuttal, we were unable to conduct detailed hyperparameter tuning to optimize our results. Therefore, these results should not be interpreted as a limitation of our model's transfer learning capability.
> >
> > On the contrary, the strong performance of our model on certain tasks highlights its robust transfer learning ability. Additionally, none of the models we benchmark against (listed in table 2 and 3) have conducted experiments on this dataset, so our results do not contradict the performance of our model on MoleculeNet.
> >
> > If the reviewer still perceives our benchmarks on downstream tasks as "lacking consistency," we kindly request clarification or specific examples. This feedback would help us identify and address any concerns.
> >
> > Lastly, as mentioned in our previous comment, we will release the code and model publicly upon acceptance of the paper.

---

### Official Review · Reviewer_37F6 · 2024-11-03

**Soundness:** 3
**Presentation:** 3
**Contribution:** 2
**Rating:** 3
**Confidence:** 5

**Summary:**

The manuscript introduces FARM (Functional Group-Aware Representations for Small Molecules), which seeks to enhance molecular representation learning by integrating functional group (FG) information into SMILES and graph-based representations. The core innovation lies in FG-aware tokenization and the use of contrastive learning to align these sequence- and graph-based molecular representations. The paper reports that FARM outperforms existing models on the MoleculeNet dataset across 10 of 12 tasks, showcasing potential in drug discovery and cheminformatics.

**Strengths:**

- The model's performance on diverse tasks from the MoleculeNet benchmark and comparisons with state-of-the-art methods provide robust evidence for its efficacy.
- The paper does an excellent job explaining the FG detection, tokenization process, and integration of representations through contrastive learning. The use of a functional group knowledge graph adds depth to the model's structure-learning capabilities.
- The tables show clear improvements over existing models in both classification and regression tasks, indicating that the incorporation of functional group information yields substantial benefits.

**Weaknesses:**

- The increase in tokenization granularity from 93 to 14,741 tokens could be seen as excessive, leading to training inefficiencies. While the authors acknowledge this, a deeper discussion on the trade-offs and potential mitigation strategies (e.g., pre-training optimizations) would enhance the paper.
- The absence of 3D information limits the model’s capacity to handle stereochemistry and spatial effects, which are crucial in many chemical tasks. The authors mention this as a future direction, but its exclusion remains a significant limitation.
- The paper briefly mentions augmentations for negative samples in contrastive learning, such as node deletion and swapping. A more detailed exploration of the impact of these strategies would provide clarity on their contribution to performance.
-  The paper does not sufficiently address the computational requirements of training FARM, given the added complexity from FG-aware tokenization and knowledge graph embeddings.
- The novelty is also a bit limited since similar methods based on functional groups have been well-explored in previous studies [1,2,3,4,5,6].

[1] Fang Y, Zhang Q, Zhang N, et al. Knowledge graph-enhanced molecular contrastive learning with functional prompt[J]. Nature Machine Intelligence, 2023, 5(5): 542-553.

[2] Sun M, Xing J, Wang H, et al. MoCL: data-driven molecular fingerprint via knowledge-aware contrastive learning from molecular graph[C]//Proceedings of the 27th ACM SIGKDD conference on knowledge discovery & data mining. 2021: 3585-3594.

[3] Xie A, Zhang Z, Guan J, et al. Self-supervised learning with chemistry-aware fragmentation for effective molecular property prediction[J]. Briefings in Bioinformatics, 2023, 24(5): bbad296.

[4] Collins E M, Raghavachari K. A fragmentation-based graph embedding framework for QM/ML[J]. The Journal of Physical Chemistry A, 2021, 125(31): 6872-6880.

[5] Wang Y, Magar R, Liang C, et al. Improving molecular contrastive learning via faulty negative mitigation and decomposed fragment contrast[J]. Journal of Chemical Information and Modeling, 2022, 62(11): 2713-2725.

[6] Kim S, Nam J, Kim J, et al. Fragment-based multi-view molecular contrastive learning[C]//Workshop on''Machine Learning for Materials''ICLR 2023. 2023.

**Questions:**

I think the main issue is the experiment section is too short without any in-depth analysis and discussion. The authors should continue adding content and polishing this section.

---

> ### Author Response · Authors · 2024-11-20
>
> We sincerely thank the reviewer for their thoughtful feedback and for highlighting key aspects of our approach. Below, we address each of the reviewers' comments in detail. The corresponding modifications in the revised manuscript are highlighted in blue for clarity.
>
> ***The increase in tokenization granularity from 93 to 14,741 tokens could be seen as excessive, leading to training inefficiencies. While the authors acknowledge this, a deeper discussion on the trade-offs and potential mitigation strategies (e.g., pre-training optimizations) would enhance the paper.***
>
> * We appreciate the reviewer’s valuable insight regarding the trade-offs associated with increased tokenization granularity. While the jump from 93 to 14,741 tokens may seem large and potentially inefficient, our findings indicate that the benefits outweigh the potential downsides.
> * In MoleBERT [1], the authors observed a negative transfer issue when using standard SMILES for masked language prediction. With only 118 tokens representing common chemical elements, the task becomes overly simplistic, failing to capture deeper, transferable knowledge. Other studies [2, 3] have shown that simpler pre-training tasks can limit a model’s ability to generalize effectively to new tasks.
>  * On the other hand, a typical BERT model for English operates with around 30K tokens, many of which are rare, yet BERT still learns effective word representations. This suggests that BERT can handle a larger vocabulary without sacrificing performance.
> * In our work, as shown in Figure 8, our FG-enhanced SMILES model with ~15K tokens achieves a loss comparable to that of models using standard SMILES with only 93 tokens, suggesting that the model can handle masked language modeling effectively with this expanded vocabulary. Moreover, Table 8 illustrates that this increased tokenization granularity significantly boosts downstream performance. Specifically, our FG-BERT model, trained on functional group-enhanced SMILES, consistently outperforms the standard SMILES-based BERT, demonstrating clear improvements in representation quality.
>
> ***The absence of 3D information limits the model’s capacity to handle stereochemistry and spatial effects, which are crucial in many chemical tasks. The authors mention this as a future direction, but its exclusion remains a significant limitation.***
>
> * We appreciate the reviewer’s insight into the importance of 3D information in molecular modeling. As noted in our paper, incorporating 3D spatial features is part of our planned future work. This study focuses on advancing functional group and molecular structure awareness within a 2D framework, which is highly effective across a range of tasks. It is worth noting that even without utilizing 3D information, our method demonstrates significant improvements on downstream tasks, as demonstrated in Table 2 and 3.
> * Our goal is to establish a robust foundation in 2D molecular representation, setting the stage for future enhancements with 3D spatial features. This stepwise progression aligns with standard scientific practice, where model extensions build logically upon established foundations. We kindly suggest that limitations related to future work not be used to detract from the contributions made in the present study.
>
> ***The paper briefly mentions augmentations for negative samples in contrastive learning, such as node deletion and swapping. A more detailed exploration of the impact of these strategies would provide clarity on their contribution to performance.***
> * Thank you for your comment on the augmentation methods used for negative samples in our contrastive learning approach. We note that node deletion and swapping are widely accepted augmentations for molecular graphs [4, 5, 6], serving to generate hard negative samples with high similarity to the original graph. These hard negative samples require the model to better distinguish between positive and negative examples, enhancing the learned features by making the model work harder to pull the FG-enhanced SMILES representation closer to the positive graph while pushing it away from the hard negative sample.
> * In the context of molecular graphs, these augmentations are analogous to standard image augmentations, like cropping or flipping, which are used without exhaustive justification in image-based models. While an in-depth analysis of each augmentation’s impact is beyond this paper’s scope, we align with prior work in selecting these strategies, as they are known to effectively improve model robustness and representation quality in molecular graph tasks.

---

> ### Author Response · Authors · 2024-11-20
>
> ***The paper does not sufficiently address the computational requirements of training FARM, given the added complexity from FG-aware tokenization and knowledge graph embeddings.***
>
> To address the reviewer’s concern about the computational requirements of training FARM with FG-aware tokenization and knowledge graph embeddings, we would like to clarify that the computational resources required for FARM are not significantly higher than those for training a standard language model (in this case is BERT model). The training process follows a multi-stage approach, where each stage is computationally manageable.
>
> Multi-stage training:
>
> * Step 1: The pre-training of the masked language model (MLM) is similar to the standard BERT pre-training procedure. The computational cost for this step is on par with that of training BERT.
> * Step 2: We construct the FG knowledge graph and train a knowledge graph embedding model, such as ComplEx [7]. Training the knowledge graph embedding model does not require additional resources compared to training a language model. Knowledge graph embedding techniques have been well established and are scalable [8].
> * Step 3: Construct the FG graphs and generate negative samples, then train the link prediction model to learn the high-level structure of the molecules, specifically how the functional groups (FGs) connect with each other. This step uses standard graph neural network techniques, such as the Graph Convolutional Network (GCN) model, and does not introduce significant computational overhead.
> * Step 4: Fine-tune the MLM model by combining contrastive learning with structure representations. This step leverages pre-trained LMs and focuses on integrating the structural information into FG-enhanced SMILES representation, which only requires fine-tuning a BERT model and a linear layer to project the structural representation into a new space, facilitating better alignment with the MLM model’s learned embeddings.
>
> Extracting molecular representations for downstream tasks: Convert the SMILES to FG-enhanced SMILES and use the pre-trained BERT model to extract molecular representations. This process does not require additional computational resources.
>
> Regarding training time, there is always a trade-off between training duration and performance. However, with only a slight increase in training time, our method achieves a substantial improvement in performance. As shown in Figure 8 (Appendix D), the loss curves for the MLM during training on two datasets—standard SMILES and FG-enhanced SMILES—demonstrate that, after 500 training steps, both losses converge to approximately the same value.

---

> ### Author Response · Authors · 2024-11-20
>
> ***The novelty is also a bit limited since similar methods based on functional groups have been well-explored in previous studies [1,2,3,4,5,6].***
>
> We acknowledge that using functional group (FG) information to improve molecular representation is not a novel concept. However, our approach is distinct in its ability to both detect functional groups and effectively incorporate functional group information to enhance molecular representations, leading to strong transfer learning capabilities, as demonstrated by performance in downstream tasks. Here’s how our work differentiates itself:
>
> * **Detecting functional groups:** Existing motif-based tokenization methods do not strictly align with chemical functional groups. For instance, BRICS-based approaches (e.g., [9, 10, 11, 12]) generate substructures that do not map precisely to known functional groups. Methods using RDKit [13, 14] can only detect a limited set of functional groups (typically 85 traditional ones). Unsupervised strategies [15] often struggle with complex ring structures. In contrast, our approach implements algorithms to detect and incorporate a comprehensive set of chemical functional groups, including ring-containing groups, ensuring that the tokenization process aligns strictly with chemical features. We discuss this in more detail in the revised version of our manuscript (Related Works section, line 212).
> * **Incorporating functional group information:** We are the first to ***directly inject functional group information into SMILES***, adding richer chemical context to SMILES, then we can effectively leverage language models (LMs) to learn molecular representations.
> * **Robustness in downstream tasks:** Our model (FARM) demonstrates ***strong transfer learning capabilities*** - the core goal of pretrained models - outperforming other methods in 10 out of 12 tasks from the MoleculeNet benchmark. This suggests that FARM effectively learns molecular representations, leading to enhanced performance across various tasks.
> * We also contribute by ***evaluating the diversity of large molecular datasets***. Identifying functional groups and embedding this information into SMILES strings enables us to build a novel vocabulary, where each token reflects both an atom symbol and its associated functional group, serving as a measure of dataset diversity. Through this approach, we find that the ZINC dataset is significantly less diverse than ChEMBL, providing valuable insight for future work on dataset selection for foundation model training. Given the vast chemical space, it is essential for training data to represent a broad range of chemical structures. Our work is the first to conduct this type of evaluation, whereas previous studies have primarily focused on molecule length and atom types, which do not fully capture functional group diversity within datasets.
>
> We have highlighted those points in the abstract and introduction of the revised manuscript.

---

> ### Author Response · Authors · 2024-11-20
>
> References
>
> [1] Jun Xia, Chengshuai Zhao, Bozhen Hu, Zhangyang Gao, Cheng Tan, Yue Liu, Siyuan Li, and Stan Z Li. Mole-bert: Rethinking pre-training graph neural networks for molecules. The Eleventh International Conference on Learning Representations, ICLR 2023, 2023.
>
> [2] Kevin Clark, Minh-Thang Luong, Quoc V. Le, and Christopher D. Manning. ELECTRA: Pretraining text encoders as discriminators rather than generators. In ICLR, 2020.
>
> [3] Joshua David Robinson, Ching-Yao Chuang, Suvrit Sra, and Stefanie Jegelka. Contrastive learning with hard negative samples. In International Conference on Learning Representations, 2021.
>
> [4] Yuning You, Tianlong Chen, Yongduo Sui, Ting Chen, Zhangyang Wang, and Yang Shen. 2020. Graph contrastive learning with augmentations. Advances in neural information processing systems, 33:5812– 5823.
>
> [5] Yuyang Wang, Jianren Wang, Zhonglin Cao, and Amir Barati Farimani. Molecular contrastive learning of representations via graph neural networks. Nature Machine Intelligence, 4(3):279– 287, 2022b.
>
> [6] Sun M, Xing J, Wang H, et al. MoCL: data-driven molecular fingerprint via knowledge-aware contrastive learning from molecular graph[C]//Proceedings of the 27th ACM SIGKDD conference on knowledge discovery & data mining. 2021: 3585-3594.
>
> [7] Trouillon, T., Welbl, J., Riedel, S., Gaussier, É., & Bouchard, G. (2016). Complex embeddings for simple link prediction. ICLR.
>
> [8] Rossi, R. A., Sardiña, S. I., & Schein, E. S. (2020). A Survey of Network Embedding Methods for Graphs and Complex Networks. ACM Computing Surveys.
>
> [9] Zaixi Zhang, Qi Liu, Hao Wang, Chengqiang Lu, and Chee-Kong Lee. Motif-based graph selfsupervised learning for molecular property prediction. Advances in Neural Information Processing Systems, 34:15870–15882, 2021.
>
> [10] Shen Han, Haitao Fu, Yuyang Wu, Ganglan Zhao, Zhenyu Song, Feng Huang, Zhongfei Zhang, Shichao Liu, and Wen Zhang. Himgnn: a novel hierarchical molecular graph representation learning framework for property prediction. Briefings in Bioinformatics, 24(5):bbad305, 2023.
>
> [11] Yongqiang Chen, Quanming Yao, Juzheng Zhang, James Cheng, and Yatao Bian. Hight: Hierarchical graph tokenization for graph-language alignment. arXiv preprint arXiv:2406.14021, 2024.
>
> [12] Nianzu Yang, Kaipeng Zeng, Qitian Wu, Xiaosong Jia, and Junchi Yan. Learning substructure invariance for out-of-distribution molecular representations. Advances in Neural Information Processing Systems, 35:12964–12978, 2022.
>
> [13] Biaoshun Li, Mujie Lin, Tiegen Chen, and Ling Wang. Fg-bert: a generalized and self-supervised functional group-based molecular representation learning framework for properties prediction. Briefings in Bioinformatics, 24(6):bbad398, 2023.
>
> [14] Yongqiang Chen, Quanming Yao, Juzheng Zhang, James Cheng, and Yatao Bian. Hight: Hierarchical graph tokenization for graph-language alignment. arXiv preprint arXiv:2406.14021, 2024.
>
> [15] Yifei Wang, Shiyang Chen, Guobin Chen, Ethan Shurberg, Hang Liu, and Pengyu Hong. Motifbased graph representation learning with application to chemical molecules. In Informatics, volume 10, pp. 8. MDPI, 2023.

---

> > ### Comment · Reviewer_37F6 · 2024-11-27
> > **Discussion**
> >
> > Thank you for the response. I have a few more questions that might need to discuss further:
> > - The argument of "larger vocab size is helpful" might be architecture-dependent. For example, there has been tons of recent research showing that with architectures like linear RNNs/attention/SSMs, for biological sequences, high resolution tokenization might be better [1,2,3]. This has even been demonstrated in natural language [5]. If this is architecture-dependent and switching to another architecture can discard the need of using larger vocab size, I don't see the motivation of using this FG-based tokenization.
> > - The argument of "other motif-based methods are not good" has not been empirically verified. If you are confident that they are less effective than your proposed approach, can you provide empirical study of them and discuss in your revised manuscript?
> >
> > [1] Sequence modeling and design from molecular to genome scale with Evo
> >
> > [2] Caduceus: Bi-Directional Equivariant Long-Range DNA Sequence Modeling
> >
> > [3] LC-PLM: Long-context Protein Language Model
> >
> > [4] A long-context RNA foundation model for predicting transcriptome architecture
> >
> > [5] MambaByte: Token-free Selective State Space Model

---

> > > ### Author Response · Authors · 2024-11-27
> > >
> > > **High-resolution tokenization**
> > >
> > > Thank you for your thoughtful response and for raising an important discussion about tokenization in biological sequences versus SMILES.
> > >
> > > You are absolutely correct that in biological sequences such as DNA, RNA, and proteins, high-resolution tokenization is an effective strategy. This approach results in tokens (letters) that inherently represent chemically meaningful units: nucleotides in DNA/RNA (A, T/U, C, G) or amino acids in proteins (e.g., A, R, N, D). These tokens carry intrinsic functional and structural significance, allowing models to interpret sequences directly and meaningfully without the need for additional preprocessing.
> > >
> > > In contrast, SMILES presents a unique challenge. Individual characters in SMILES (e.g., C, O, N) do not correspond to functional chemical units like nucleotides or amino acids in biomolecules. Instead, functional groups in small molecules better align with the concept of amino acids or nucleotides in biomolecules, as they represent distinct substructures with specific chemical properties. High-resolution tokenization in SMILES, which uses characters as tokens, often results in tokens dominated by carbon atoms. As highlighted in the MoleBERT paper [1], this approach over-simplifies tasks like masked language modeling (MLM), limits the model’s ability to learn meaningful features, and leads to negative transfer effects on downstream tasks. Recognizing this limitation, many works have focused on building block-based molecular tokenization to capture more chemically meaningful substructures [2, 3, 4, 5, 6, 7]. Here, we aim to inject functional group information into SMILES strings to provide richer chemical context and to enhance the difficulty of the MLM task, encouraging the model to learn more meaningful and robust features from the data.
> > >
> > > **Ablation study of tokenization methods**
> > >
> > > Regarding the ablation study comparing our functional group-aware tokenization with other motif-based tokenization methods, we acknowledge that we have not provided this comparison in the current paper. Due to the limited time available during the rebuttal period, we were unable to conduct such an experiment.
> > >
> > > What we aim to convey is that our approach focuses on tokenizing molecules into ***well-defined functional groups*** (such as those listed on this [Wikipedia page](https://en.wikipedia.org/wiki/Functional_group)), as these groups have a significant impact on molecular properties. We are not claiming that *"other motif-based methods are not good."* Each method may have its strengths, but our primary goal is to capture as much molecular functional information as possible to enrich the SMILES representation. By tokenizing molecules into functional groups—considered the functional units of the molecules—we aim to provide SMILES with additional information about the molecular functions, improving its ability to represent chemical behavior more accurately.
> > >
> > > [1] Jun Xia, Chengshuai Zhao, Bozhen Hu, Zhangyang Gao, Cheng Tan, Yue Liu, Siyuan Li, and
> > > Stan Z Li. Mole-bert: Rethinking pre-training graph neural networks for molecules. The Eleventh
> > > International Conference on Learning Representations, ICLR 2023, 2023.
> > >
> > > [2] Shichang Zhang, Ziniu Hu, Arjun Subramonian, and Yizhou Sun. Motif-driven contrastive learning
> > > of graph representations. arXiv preprint arXiv:2012.12533, 2020.
> > >
> > > [3] Zaixi Zhang, Qi Liu, Hao Wang, Chengqiang Lu, and Chee-Kong Lee. Motif-based graph selfsupervised learning for molecular property prediction. Advances in Neural Information Processing Systems, 34:15870–15882, 2021.
> > >
> > > [4] Nianzu Yang, Kaipeng Zeng, Qitian Wu, Xiaosong Jia, and Junchi Yan. Learning substructure invariance for out-of-distribution molecular representations. Advances in Neural Information Processing Systems, 35:12964–12978, 2022.
> > >
> > > [5] Shen Han, Haitao Fu, Yuyang Wu, Ganglan Zhao, Zhenyu Song, Feng Huang, Zhongfei Zhang,
> > > Shichao Liu, and Wen Zhang. Himgnn: a novel hierarchical molecular graph representation
> > > learning framework for property prediction. Briefings in Bioinformatics, 24(5):bbad305, 2023.
> > >
> > > [6] Biaoshun Li, Mujie Lin, Tiegen Chen, and Ling Wang. Fg-bert: a generalized and self-supervised
> > > functional group-based molecular representation learning framework for properties prediction.
> > > Briefings in Bioinformatics, 24(6):bbad398, 2023.
> > >
> > > [7] Xuan Zang, Xianbing Zhao, and Buzhou Tang. Hierarchical molecular graph self-supervised learning for property prediction. Communications Chemistry, 6(1):34, 2023.

---

> > > > ### Comment · Reviewer_37F6 · 2024-11-27
> > > > **Discussion**
> > > >
> > > > Thanks for the reply.
> > > > - The discussion period is Nov.13rd to Dec.3rd during which you have more than 20 days to provide additional results. I don't think "the limited time" is a reasonable excuse to avoid putting out a fair comparison with other motif-based tokenization methods.
> > > > - I don't buy your idea of having well-defined functional groups as tokens. For example, in natural language, we don't tokenize text into words; we instead tokenize them into sub-words or so. Thus, your claims like "For instance, BRICS-based approaches (e.g., [9, 10, 11, 12]) generate substructures that do not map precisely to known functional groups." couldn't hold without empirical evidence.

---

> > > > > ### Author Response · Authors · 2024-11-27
> > > > >
> > > > > We acknowledge the importance of providing a comparison with other motif-based tokenization methods. However, our FARM training process took approximately 20 days to train on 20 million SMILES strings over 18 epochs. To ensure a fair comparison, we would need to replicate the same training process, which is time-intensive. Additionally, testing multiple models with various tokenization methods would further extend the time required.
> > > > >
> > > > > We also want to kindly remind you that the request for this ablation study was made today, rather than in your initial review.

---

> > > > > > ### Comment · Reviewer_37F6 · 2024-11-27
> > > > > > **Discussion**
> > > > > >
> > > > > > You can train on a smaller subset of data and less epochs as long as you keep the setting consistent for different model variants. This is enough for ablation studies. Also, I raised my concern about this tokenization thing in my initial review. You claimed something only based on your assumption which lacks evidence. I was just saying that you have to provide empirical evidence if you want to claim it. Otherwise, this concern couldn’t be addressed.

---

> > > > > > > ### Author Response · Authors · 2024-11-27
> > > > > > >
> > > > > > > Thank you for your quick response. In your original review, the only comment on our tokenization method was that *"the novelty is a bit limited since similar methods based on functional groups have been well-explored in previous studies."* We addressed this by pointing out the novelty of our work. However, there was no specific request or suggestion to conduct an ablation study comparing our method with other tokenization methods in your initial review. If your original review had been more specific and constructive, we would have known to include such a comparison. We only acknowledged the request for the ablation study after your first response.

---

> > > > > > > > ### Comment · Reviewer_37F6 · 2024-11-27
> > > > > > > > **Discussion**
> > > > > > > >
> > > > > > > > Thanks for the reply. Given my concerns still remain, I’ll keep the score as is.

---

### Meta-Review · Area_Chair_eFbx · 2024-12-21

**Metareview:**

This paper proposes a new functional-group representation of molecules, which improve upon the existing representations (including SMILES) by incorporating prior knowledge of functional groups.

I think the idea is interesting and provides conceptual improvement over the existing fragment- or motif-based approaches to represent molecules. However, the reviewers were not convinced by the method's novelty and improvement, particularly because the paper does not include direct comparisons or ablation studies with respect to the existing works that use fragment- or motif-based approaches.

Overall, I recommen rejection since the concerns are not fully resolved. This could be a strong submission for the next conference if the authors make a strong empirical evaluation to resolve this issue.

**Additional Comments On Reviewer Discussion:**

The reviewers were responsive during the rebuttal. I acknowledge that reviewer 37F6 mentioned the necessity of time-consuming experiment near the end of rebuttal phase. However, while it is unfortunate that this comment has been brought up too late, this is not the reviewer's fault. This was just a suggestion of reviewer 37F6 as a way of alleviating the concerns. I agree with reviewer 37F6 that the comparison to existing motif-based tokenization methods is necessary for the paper to get accepted.

---

### Decision · Program_Chairs · 2025-01-22

Reject